# Function-Space Learning Rates

**Edward Milsom** [1]   **Ben Anson** [1]   **Laurence Aitchison** [1]

## Abstract

We consider layerwise function-space learning rates, which measure the magnitude of the change in a neural network's output function in response to an update to a parameter tensor. This contrasts with traditional learning rates, which describe the magnitude of changes in parameter space. We develop efficient methods to measure and set function-space learning rates in arbitrary neural networks, requiring only minimal computational overhead through a few additional backward passes that can be performed at the start of, or periodically during, training. We demonstrate two key applications: (1) analysing the dynamics of standard neural network optimisers in function space, rather than parameter space, and (2) introducing FLeRM (Function-space Learning Rate Matching), a novel approach to hyperparameter transfer across model scales. FLeRM records function-space learning rates while training a small, cheap base model, then automatically adjusts parameter-space layerwise learning rates when training larger models to maintain consistent function-space updates. FLeRM gives hyperparameter transfer across model width, depth, initialisation scale, and LoRA rank in various architectures including MLPs with residual connections and transformers with different layer normalisation schemes.

## 1. Introduction

The fundamental purpose of neural network training is to learn a function that maps inputs to desired outputs. However, we typically understand optimisation methods as acting in parameter space, e.g. traditional learning rates tell us how much the parameters change during each step rather than the functional impact of those changes. This raises an important question: can we meaningfully quantify and control learning in function space?

We consider the concept of layerwise function-space learning rates which measure the magnitude of change in network output induced by updates to individual parameter tensors. Unfortunately, naive approaches to measuring function-space learning rates would be computationally prohibitive. We solve this problem by developing a Monte-Carlo estimate to measure function-space learning rates using only a single additional backward pass, which can be performed a handful of times at the start of training (e.g. 40), or periodically during training (e.g., once every 100 steps), resulting in negligible computational overhead. We then consider two immediate applications of function-space learning rates.

First, function-space learning rates provide a novel lens for analysing the behavior of standard neural network optimisers (e.g. Adam, Kingma, 2014), giving important insights into how different parts of the network contribute to functional changes during training. Second, function-space learning rates enable a new approach to hyperparameter transfer (for previous work see e.g. Yang & Hu, 2022; Bordelon et al., 2023; Large et al., 2024). As large language model pre-training can cost millions of dollars in compute (Cottier et al., 2024), running extensive hyperparameter sweeps at full scale is impractical. Instead, one might hope to optimise hyperparameters on smaller models and transfer them to larger ones. However, this is complicated by the fact that optimal learning rates change with model width and depth (Yang & Hu, 2022; Noci et al., 2024; Yang et al., 2023; Bordelon et al., 2023). We address this challenge with FLeRM (Function-space Learning Rate Matching), which maintains consistent function-space learning rates as models are scaled up by automatically adjusting the parameter-space learning rates, thereby keeping the optimal value of the user-defined learning rate hyperparameter stable.

A key advantage of our approach is its flexibility: our methods for measuring and controlling function-space learning rates work with any network architecture and at any point during training. This contrasts with traditional approaches to hyperparameter transfer that often make restrictive assumptions about architectures or initialisation schemes (Everett et al., 2024; Yang & Hu, 2022; Bordelon et al., 2023; Large et al., 2024). We demonstrate FLeRM's utility across a

---

[1]University of Bristol. Correspondence to: Edward Milsom <edward.milsom@bristol.ac.uk>, Laurence Aitchison <laurence.aitchison@gmail.com>.

*Proceedings of the $42^{nd}$ International Conference on Machine Learning*, Vancouver, Canada. PMLR 267, 2025. Copyright 2025 by the author(s).

range of scenarios, including model width scaling, depth scaling, initialisation scale variation, and even LoRA rank adjustment.

## 2. Related work

As far as we are aware, this work is unique in proposing methods to measure and set function-space learning rates in arbitrary neural networks far from initialisation, and using this approach to study the dynamics of optimizers.

There is an existing body of literature on hyperparameter transfer (e.g. Yang & Hu, 2022; Bordelon et al., 2023; Large et al., 2024; Yaida, 2022). Broadly speaking, these works analytically derive scaling laws for e.g. the initialisations and parameter-space learning rates, such that the function-space learning rates do not change as e.g. width is increased. This is radically different from our approach to hyperparameter scaling. In particular, none of these works provide a mechanism to empirically measure the function-space learning rates in an arbitrary neural network. Furthermore, prior works tend to rely on rigid assumptions such as being close to a specific random initialisation, which we do not make.

Perhaps the earliest and best-known work on hyperparameter scaling is μP (Yang & Hu, 2022). μP derives how to scale random initialisations and learning rates as you increase model width, such that the function-space learning rates remain asymptotically constant (i.e. the magnitude of the activations does not blow up to infinity or shrink to zero as the model gets wider). μP has since been extended to depth-scaling (Yang et al., 2023; Bordelon et al., 2023) and networks trained with sharpness-aware minimisation (Haas et al., 2024), and is closely related to the mean-field analysis of neural networks grounded in statistical mechanics (Mei et al., 2018; Rotskoff & Vanden-Eijnden, 2022; Sirignano & Spiliopoulos, 2019; Chizat & Bach, 2018; Geiger et al., 2020; Bordelon & Pehlevan, 2022). Because this approach derives the function-space learning rates analytically, (in contrast to our approach of measuring them), it requires restrictive assumptions, including that the network is wide, and close to a random initialisation. Extending these results to more general cases is made complicated by the fact that they typically rely on heavy-duty mathematical machinery such as Tensor-Programs (Yang, 2021b; 2020; 2021a; Yang & Hu, 2022; Yang et al., 2022; 2023) or dynamical mean-field theory (Bordelon & Pehlevan, 2022; 2023; Bordelon et al., 2023; 2024).

The full μP scheme can be complex to apply in practice, because it e.g. requires distinct treatment of the initialisation and learning rates for the embedding weights and output heads. Later work (Large et al., 2024) sought to address this issue, by providing a library (Modula) of modules, such that when the modules are combined, the overall network nat-

urally exhibits hyperparameter scaling. Rather than studying scaling asymptotically, Modula follows a metrisation-based approach (for other metrisation works, see e.g. Yang et al., 2024; Bernstein et al., 2021; Bernstein & Newhouse, 2024a;b). Its carefully designed modules allow the computation of the network's Lipschitz constant, which can be used to normalise updates to enable hyperparameter scaling. However, one important difficulty with Modula is that it requires setting the "mass" of each parameter. This mass can be seen as analogous to the layerwise function-space learning rate in our work, for the first step of the optimiser (i.e. at initialisation). This introduces a large number of new hyperparameters that must be tuned. In contrast, our approach to hyperparameter transfer, FLeRM, does not require the user to specify masses / function-space learning rates for each parameter, because it directly measures the function-space learning rates in a base model, and then uses them in a scaled model. We show in Section 4.2.5 that using function-space learning rates measured directly from a base model leads to better training loss than simplistic user-defined function-space learning rates. Furthermore, FLeRM can be applied to any existing neural network in Pytorch, whereas Modula requires the user to rewrite their network architecture using the library's modules.

Chizat & Netrapalli (2024) quantify feature learning in neural networks as the angle between feature updates and backward passes, enabling analysis of hyperparameter scaling laws and development of improved scaling rules for deep networks. By contrast, our work directly measures the function-space learning rates using autodiff and a Monte-Carlo approximation.

Finally, Everett et al. (2024) recently showed empirically that alignment (concerning the size of dot products between activations and updates across different layers) in real models is highly dynamic and complex throughout training. This can make choosing the correct alignment assumptions in μP and mean-field parametrisations (Yang & Hu, 2022; Mei et al., 2018; Bordelon & Pehlevan, 2022), a very difficult task. By contrast, we propose methods that can directly measure the function-space learning rates throughout training, avoiding the need for such assumptions and analysis.

## 3. Methods

In Section 3.1 we describe how we can empirically measure the layerwise function-space learning rates using a Monte-Carlo estimate. Then, in Section 3.2, we propose using Kronecker factorisation to reduce the variance of our estimates. Finally, in Section 3.3, we introduce FLeRM, which modifies the parameter-space learning rates of scaled models so their function-space learning rates match a small base model, thereby enabling hyperparameter transfer.

## 3.1. Monte-Carlo estimation of the layerwise function-space learning rate

At the core of our contributions is the estimation of the layerwise function-space learning rates, i.e. the magnitude of the change in output logits arising from a particular change in the $\ell^{\text{th}}$ parameter tensor. We begin by considering the full change in the function output, $f_{nk}$. Here, $n$ indexes the $N$ datapoints in the minibatch, and $k$ indexes the $K$ output features. We use $\Delta_\ell f_{nk}$ to denote a first-order Taylor approximation of the change in the outputs due to a particular change, $\Delta \mathbf{W}^\ell$, in the $\ell$th parameter, $\mathbf{W}^\ell \in \mathbb{R}^{N_\ell \times N_{\ell-1}}$,

$$\Delta_\ell f_{nk} = \sum_{ij} \Delta W_{ij}^\ell \frac{df_{nk}}{dW_{ij}^\ell}. \tag{1}$$

Here, $W_{ij}^\ell$ is the $ij$th element of $\mathbf{W}^\ell$ and $\frac{df_{nk}}{dW_{ij}^\ell}$ is the gradient of the output w.r.t. $W_{ij}^\ell$. Note that for ease of notation we assume $\mathbf{W}^\ell$ is a matrix, but $\mathbf{W}^\ell$ can be a tensor of any rank. We are interested in the **layerwise function-space learning rate**, i.e. the RMS norm of $\Delta_\ell \mathbf{f}$,

$$\|\Delta_\ell \mathbf{f}\|_{\text{RMS}}^2 = \tfrac{1}{NK} \sum_{nk} (\Delta_\ell f_{nk})^2, \tag{2}$$

(L2-norm can be used if preferred, the only difference is the term $\frac{1}{NK}$). Naïvely computing Eq. (2) via Eq. (1) is intractable as it requires $NK$ backward passes, one for each $\frac{df_{nk}}{dW_{ij}^\ell}$. Instead, we exploit the fact that we only need the magnitude $\|\Delta_\ell \mathbf{f}\|_{\text{RMS}}^2$, not the full change $\Delta \mathbf{f}$. Specifically, we use a Monte-Carlo approach. Consider the following scaled random combination of outputs,

$$\phi = \tfrac{1}{\sqrt{NK}} \sum_{nk} \omega_{nk} f_{nk} \qquad \omega_{nk} \sim \mathcal{N}(0,1). \tag{3}$$

As in Eq. (1) we can write the change in $\phi$ arising from a change in the $\ell$th parameter,

$$\Delta_\ell \phi = \sum_{ij} \Delta W_{ij}^\ell \frac{d\phi}{dW_{ij}^\ell}. \tag{4}$$

Importantly, note that we can compute $\Delta_\ell \phi$ in a single backward pass. To see how $\Delta_\ell \phi$ helps us compute $\|\Delta_\ell \mathbf{f}\|_{\text{RMS}}$, we substitute the definition of $\phi$ (Eq. 3) into Eq. (4),

$$\Delta_\ell \phi = \sum_{nk} \omega_{nk} \tfrac{1}{\sqrt{NK}} \sum_{ij} \Delta W_{ij}^\ell \frac{df_{nk}}{dW_{ij}^\ell} \tag{5}$$

and note that the inner sum is $\Delta_\ell f_{nk}$ (Eq. 1), so simplifying,

$$\Delta_\ell \phi = \sum_{nk} \omega_{nk} \tfrac{1}{\sqrt{NK}} \Delta_\ell f_{nk}, \tag{6}$$

and since $\omega_{nk}$ are IID standard Gaussian (Eq. 3), we have

$$\Delta_\ell \phi \sim \mathcal{N} \left(0, \|\Delta_\ell \mathbf{f}\|_{\text{RMS}}^2\right). \tag{7}$$

Hence we can estimate $\|\Delta_\ell \mathbf{f}\|_{\text{RMS}}^2$ by computing $\Delta_\ell \phi$ with multiple samples of $\omega_{nk}$, and estimating the variance.

**Algorithm 1** Recording (red) or setting (FLeRM, blue) function-space learning rates in a training loop.

---

**Input:** $\|\Delta_\ell \mathbf{f}\|_{\text{RMS}}^{(\text{base},:)}$    Base model function-space LRs
$\texttt{EMA\_Z2}[\ell], \texttt{EMA\_ZZT}[\ell], \texttt{EMA\_ZTZ}[\ell] = 0$
**for** $t = 1$ **to** $T$ **do**
  $\mathbf{f} \leftarrow f(\mathbf{X})$                  Std. forward pass
  $\mathcal{L} \leftarrow \text{loss}(\text{targets}, \mathbf{f})$           Std. loss
  $g_{ij}^\ell \leftarrow d\mathcal{L}/dW_{ij}^\ell$        Std. backward pass
  $\mathbf{W}_{\text{buffer}}^\ell \leftarrow \mathbf{W}^\ell$     Save weights before update
  $\mathbf{W}^\ell \leftarrow \text{optimiser}(\eta_0, \mathbf{W}^\ell, g^\ell)$   Std. update (base LR)
  $\mathbf{W}^\ell \leftarrow \text{optimiser}(\{\eta^\ell\}_{\ell=1}^L, \mathbf{W}^\ell, g^\ell)$ (or FLeRM LR)
  EMA warmup: run below code for a few different $\mathbf{X}'$
  **if** t % 100 == 0 **then**
    $\omega_{nk} \sim \mathcal{N}(0,1)$
    $\mathbf{f}' = f(\mathbf{X}')$          Fresh batch of data for $\phi$
    $\phi \leftarrow \tfrac{1}{\sqrt{NK}} \sum_{nk} \omega_{nk} f_{nk}'$     Compute $\phi$ (Eq. 3)
    $g_{ij}^\ell \leftarrow d\phi/dW_{ij}^\ell$        Backward pass for $\phi$.
    $\Delta \mathbf{W}^\ell \leftarrow \tfrac{1}{\eta^\ell} \left(\mathbf{W}^\ell - \mathbf{W}_{\text{buffer}}^\ell\right)$      LR=1 update
    $Z_{ij}^\ell \leftarrow g_{ij}^\ell \Delta W_{ij}^\ell$     Compute $Z_{ij}$ as in Eq. (8)
    Update EMAs
    $\texttt{EMA\_Z2}[\ell] \leftarrow (1-\beta)\texttt{EMA\_Z2}[\ell] + \beta \sum_{ij} Z_{ij}^2$
    $\texttt{EMA\_ZZT}[\ell] \leftarrow (1-\beta)\texttt{EMA\_ZZT}[\ell] + \beta \sum_k (\sum_i Z_{ik})^2$
    $\texttt{EMA\_ZTZ}[\ell] \leftarrow (1-\beta)\texttt{EMA\_ZTZ}[\ell] + \beta \sum_k (\sum_j Z_{kj})^2$
    Function-space LR (EMA bias correction hidden)
    $\|\Delta_\ell \mathbf{f}\|_{\text{RMS}}^{(t)} \leftarrow \sqrt{\texttt{EMA\_ZZT}[\ell]\texttt{EMA\_ZTZ}[\ell]/\texttt{EMA\_Z2}[\ell]}$
    Set parameter-space learning rates (FLeRM)
    $\eta^\ell \leftarrow \eta_0 \|\Delta_\ell \mathbf{f}\|_{\text{RMS}}^{(\text{base},t)} / \|\Delta_\ell \mathbf{f}\|_{\text{RMS}}^{(t)}$
  **end if**
**end for**
**Output:** $\|\Delta_\ell \mathbf{f}\|_{\text{RMS}}^{(:)}$ Recorded function-space LRs

---

## 3.2. More efficient function-space learning rate estimates using Kronecker factorisation

The approach in Sec. 3.1 requires one backward pass per sample, which could still be inefficient if we need multiple samples for a good estimate. We remedy this by exploiting the structure of $\phi$. Noting that $i, j$ index the rows and columns of $\mathbf{W}^\ell$, we rewrite Eq. (4) using $\mathbf{Z} \in \mathbb{R}^{N_\ell \times N_{\ell-1}}$,

$$\Delta_\ell \phi = \sum_{ij} Z_{ij} \qquad Z_{ij} = \Delta W_{ij}^\ell \frac{d\phi}{dW_{ij}^\ell} \tag{8}$$

Hence the function-space learning rate can be written as

$$\|\Delta_\ell \mathbf{f}\|_{\text{RMS}}^2 = \text{Var}[\Delta_\ell \phi] = \sum_{ij,i'j'} \text{Cov}[Z_{ij}, Z_{i'j'}]. \tag{9}$$

Also note that, substituting the definition of $\phi$ (Eq. 3) into the definition of $Z_{ij}$ (Eq. 8), we see that the $Z_{ij}$'s are zero-mean Gaussian, as they are a linear combination of zero-

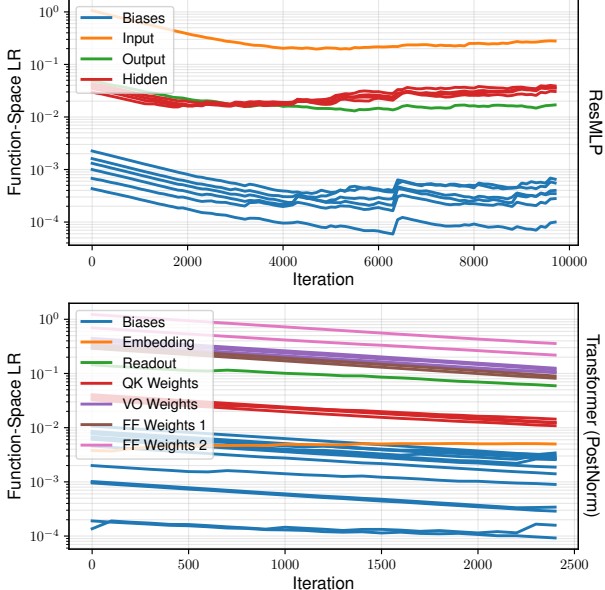

*Figure 1.* Function-space learning rates over time, measured using our approach, for the ResMLP model (top) and the Transformer (PostNorm) model (bottom). "QK Weights" refers to $W_Q$ or $W_K$ (query and key weight matrices), whilst "VO Weights" refers to $W_V$ or $W_O$ (values and head-concatenation projection weight matrices).

mean Gaussian terms $\omega_{nk}$ (we will use this fact later),

$$Z_{ij} = \sum_{nk} \omega_{nk} \frac{1}{\sqrt{NK}} \Delta W_{ij}^{\ell} \frac{df_{nk}}{dW_{ij}^{\ell}}. \quad (10)$$

From Eq. (9) we can see that instead of directly estimating the variance of samples of $\Delta_\ell \phi$, we could instead first estimate the covariance of the $Z_{ij}$'s and sum over them. This opens up the possibility of assuming some covariance structure over the $Z_{ij}$'s to reduce the variance of our estimate for $\text{Var}\left[\Delta_\ell \phi\right]$, at the expense of some bias. One possibly restrictive example is to assume that all the $Z_{ij}$'s are IID, in which case $\text{Var}\left[\Delta_\ell \phi\right] = \sum_{ij} \text{Var}\left[Z_{ij}\right]$. Instead, we assume a Kronecker-factored covariance matrix (Martens & Grosse, 2015). Specifically, we assume that we have a covariance matrix $\mathbf{U} \in \mathbb{R}^{N_\ell \times N_\ell}$ over rows and $\mathbf{V} \in \mathbb{R}^{N_{\ell-1} \times N_{\ell-1}}$ over columns, giving

$$\text{Cov}\left[Z_{ij}, Z_{i'j'}\right] = U_{ii'}V_{jj'}. \quad (11)$$

Under this assumption, Eq. (9) becomes

$$\begin{aligned} \|\Delta_\ell \mathbf{f}\|_{\text{RMS}}^2 &= \sum_{ij,i'j'} \text{Cov}\left[Z_{ij}, Z_{i',j'}\right] \\ &= \sum_{ij,i'j'} U_{ii'}V_{jj'} \\ &= \left(\sum_{ii'} U_{ii'}\right)\left(\sum_{jj'} V_{jj'}\right). \end{aligned} \quad (12)$$

We will now show how to compute this efficiently from $\mathbf{Z}$, which itself can be computed with a single backwards pass (Eq. 8). Since the $Z_{ij}$'s are zero-mean Gaussian, and we have assumed $\text{Cov}\left[Z_{ij}, Z_{i'j'}\right] = U_{ii'}V_{jj'}$, $\mathbf{Z}$ is zero-mean Matrix-Normal distributed, and so (Gupta & Nagar, 1999)

$$\mathbb{E}[\mathbf{Z}\mathbf{Z}^T] = \mathbf{U}\text{tr}(\mathbf{V}) \quad (13a)$$

$$\mathbb{E}[\mathbf{Z}^T\mathbf{Z}] = \mathbf{V}\text{tr}(\mathbf{U}). \quad (13b)$$

Dividing Eq. (13a) by $\text{tr}(\mathbf{V})$ and Eq. (13b) by $\text{tr}(\mathbf{U})$, we obtain $\mathbf{U}$ and $\mathbf{V}$ and can substitute them into Eq. (12),

$$\|\Delta_\ell \mathbf{f}\|_{\text{RMS}}^2 = \frac{\left(\sum_{ii'} \mathbb{E}[\mathbf{Z}\mathbf{Z}^T]_{ii'}\right)\left(\sum_{jj'} \mathbb{E}[\mathbf{Z}^T\mathbf{Z}]_{jj'}\right)}{\text{tr}(\mathbf{U})\text{tr}(\mathbf{V})} \quad (14)$$

To obtain the denominator, we take the trace of both sides of Eq. (13a) or Eq. (13b), giving us

$$\begin{aligned} \text{tr}(\mathbf{U})\text{tr}(\mathbf{V}) &= \text{tr}\left(\mathbb{E}[\mathbf{Z}\mathbf{Z}^T]\right) \quad (15) \\ &= \mathbb{E}\left[\sum_{ij} Z_{ij}^2\right] = \mathbb{E}\left[\|\mathbf{Z}\|_{\mathcal{F}}^2\right]. \end{aligned}$$

Substituting this into Eq. (14), we obtain

$$\|\Delta_\ell \mathbf{f}\|_{\text{RMS}}^2 = \frac{\mathbb{E}\left(\sum_{ii'}[\mathbf{Z}\mathbf{Z}^T]_{ii'}\right)\mathbb{E}\left(\sum_{jj'}[\mathbf{Z}^T\mathbf{Z}]_{jj'}\right)}{\mathbb{E}[\|\mathbf{Z}\|_{\mathcal{F}}^2]} \quad (16)$$

and we note that the sums in the numerator can be computed in quadratic time (not cubic as $\mathbf{Z}\mathbf{Z}^T$ or $\mathbf{Z}^T\mathbf{Z}$ suggests), e.g.

$$\begin{aligned} \sum_{ii'}[\mathbf{Z}\mathbf{Z}^T]_{ii'} &= \sum_{ii'k} Z_{ik}Z_{i'k} \\ &= \sum_k \left(\sum_i Z_{ik}\right)\left(\sum_{i'} Z_{i'k}\right) \\ &= \sum_k \left(\sum_i Z_{ik}\right)^2. \quad (17) \end{aligned}$$

Hence to compute the layerwise function-space learning rate $\|\Delta_\ell \mathbf{f}\|_{\text{RMS}}^2$, we only need to estimate 3 scalar expectations.

$$\mathbb{E}[\|\mathbf{Z}\|_{\mathcal{F}}^2], \quad \mathbb{E}[\sum_k (\sum_i Z_{ik})^2], \quad \mathbb{E}[\sum_k (\sum_j Z_{kj})^2] \quad (18)$$

Since we usually measure $\|\Delta_\ell \mathbf{f}\|_{\text{RMS}}^2$ as it changes over time, we estimate these expectations using exponential moving averages (EMAs) to further reduce the variance.

This approach can be generalised to tensor-valued parameters (see Appendix B); the resulting algorithm has very similar computational cost and memory; in particular, we only need to store $R + 1$ scalar EMAs for each parameter tensor, where $R$ is the rank of the tensor (e.g. 2 for a matrix). In summary (see Algorithm 1, red text), estimating the layerwise function-space learning rates involves computing a sample $\mathbf{Z}$ (Eq. 8), which requires a single forward pass and backward pass using a fresh batch of data to compute $\frac{d\phi}{dW_{ij}^{\ell}}$ (in addition to the usual training step which gives $\Delta W_{ij}^{\ell}$), and updating 3 scalar EMAs (for matrix parameters). We warm up the EMAs by computing several samples of $\mathbf{Z}$ at the start of training, and only use one sample of $\mathbf{Z}$ every e.g. $100^{\text{th}}$ iteration thereafter.

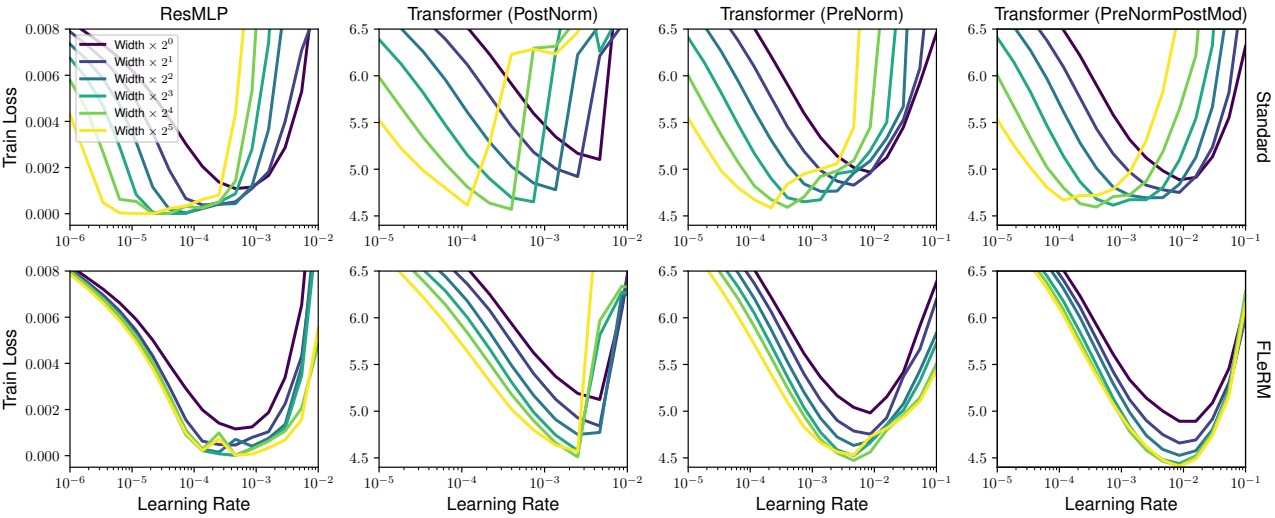

*Figure 2.* FLeRM dramatically improves optimal learning rate transfer across widths. Top: standard practice. Bottom: FLeRM.

### 3.3. Function-space learning rate matching (FLeRM)

FLeRM uses the machinery developed above to record the function-space learning rates $\|\Delta_\ell \mathbf{f}\|_{\text{RMS}}^{(\text{base},t)}$ at iteration $t$ in a small, cheap base model. Then, in the larger, more expensive model, FLeRM uses the ratio between the current function-space learning rates $\|\Delta_\ell \mathbf{f}\|_{\text{RMS}}^{(t)}$ and the base model function-space learning rates $\|\Delta_\ell \mathbf{f}\|_{\text{RMS}}^{(\text{base},t)}$ to set the parameter-space learning rates at time $t$, such that the function-space learning rates match those in the base model (see Alg. 1, blue text, for details). Note that for efficiency reasons, this is usually done at regular intervals or once at the start of training, rather than at every iteration.

There are two extra details worth discussing. First, recall that we use EMAs for the three scalars in Eq. (18). These EMA estimates will have considerable bias if the parameter-space learning rates vary (e.g. due to scheduling, or FLeRM), as previous updates of the EMA may have used a very different learning rate. Hence, in Algorithm (1), we always consider a learning rate of 1 when computing $\|\Delta_\ell \mathbf{f}\|_{\text{RMS}}$, rather than the actual learning rates. Computing the $\Delta \mathbf{W}^\ell$ implied by a learning rate of 1 ensures that the learning rate seen by the EMA is always consistent. This also applies when recording the base function-space learning rates $\|\Delta_\ell \mathbf{f}\|_{\text{RMS}}^{\text{base}}$, so the modified layerwise learning rate $\eta^\ell$ becomes

$$\eta^\ell = \frac{\eta_0 \, \|\Delta_\ell \mathbf{f}\|_{\text{RMS}}^{\text{base}}}{\|\Delta_\ell \mathbf{f}\|_{\text{RMS}}} \quad (19)$$

where $\eta_0$ is the learning rate of the base model.

Second, if the scaled model has more layers than the base model, it is not possible to "match" the layerwise function-space learning rates one-to-one. Since we scale depth by increasing the number of "residual blocks" in our experiments,

we use the heuristic of sharing the base model's function-space learning rates between the new blocks. For example, if a parameter in the first residual block has $\|\Delta_\ell \mathbf{f}\|_{\text{RMS}}^{(\text{base})} = 1$ in the base model, and the scaled model has $4\times$ as many blocks, the corresponding parameters in the first 4 blocks of the scaled model will use a base function-space learning rate of $\|\Delta_\ell \mathbf{f}\|_{\text{RMS}}^{(\text{base})} = \frac{1}{4}$.

## 4. Experiments

In this section we analyse function-space learning rates for concrete neural networks (Section 4.1), and investigate the use of FLeRM to enable hyperparameter transfer when scaling model width, depth, initialisation scale, and LoRA rank (Section 4.2).

Full details of the models[1] used in the experiments can be found in Appendix A. The base ResMLP is an MLP with 4 hidden layers, each with residual connections, trained for 50 epochs on flattened CIFAR-10 images (Krizhevsky & Hinton, 2009). The base transformer is decoder-only, has two self-attention + feedforward blocks (Vaswani et al., 2017), and is trained on a subset of the Wikitext-103 dataset (Merity et al., 2016). The widest transformer has roughly 814M parameters. We compare 3 different types of Layernorm (Ba et al., 2016) in the transformers, with affine transformations disabled: PostNorm is $\text{Norm}(x + f(x))$, PreNorm is $x + f(\text{Norm}(x))$, and PreNormPostMod is $x + \text{Norm}(f(x))$. In the ResMLP, scaling width increases the hidden dimension, whilst in the transformers, we scale the embedding dimension, the feedforward hidden dimension, and the num-

---

[1]We provide our code at https://github.com/edwardmilsom/function-space-learning-rates-paper

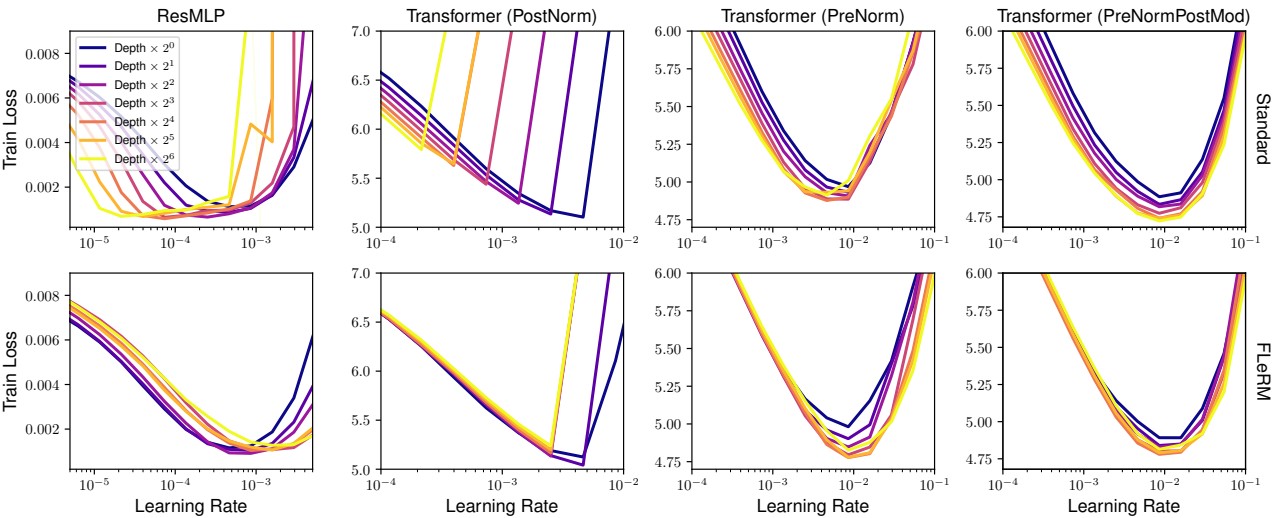

*Figure 3.* FLeRM improves or maintains optimal learning rate transfer across depth. Top: standard practice. Bottom: (FLeRM).

ber of heads. In all models, depth scaling increases the number of "residual blocks" that form the hidden layers of the model. Both ResMLP and the transformers used the Adam optimiser (Kingma, 2014) with a constant learning rate schedule. In all plots, train loss is averaged over the last 200 batches of training.

### 4.1. Analysing function-space learning rates

In Figure 1 we measure the function-space learning rates using the techniques presented in Sections 3.1 and 3.2 for the ResMLP and PostNorm transformer models. Plots for the PreNorm and PreNormPostMod transformer can be found in Figure 6, though they are very similar. We use 40 batches of data to warm up the EMAs as suggested in Section 3.2, then measure the function-space learning rate every 100 iterations. We use an EMA decay rate of $\beta = 0.9$.

In Figure 1 we see that in both models, the function-space learning rates change over time, despite the parameter-space learning rates being fixed. The most obvious pattern is that the function space learning rates fall monotonically for all parameters, except the input embedding layer, revealing an implicit scheduling of the function-space learning rates under standard Adam training. Interestingly, in the ResMLP, whilst the function-space learning rates initially decay, the hidden layers and input layer eventually start increasing again, whilst the output layer plateaus. In the transformer, the embedding layer's function-space learning rate actually increases over time. One possible explanation is that the noisy initialisation in hidden and output layers effectively scrambles the signal from the input layer, but as these layers are trained, the input embedding can have a clearer, stronger effect on the output.

We also observe that the different types of layers, such as feedforward weights or QK weights, form very clear "groups" or "bands" in these plots. From this, we can see that the second feedforward weight matrices in self-attention have the strongest influence over the transformer's learned function, with function-space learning rates an order of magnitude larger than those of the readout layer. This is a surprising discovery, since one might naively expect the readout layer, whose weights directly project to the output logits, to have the largest effect on the learned function.

### 4.2. FLeRM hyperparameter transfer experiments

We evaluated the effect of FLeRM on hyperparameter transfer when scaling model width, depth, parameter initialisation scale, and LoRA rank. We ensure width invariance at initialisation by using Kaiming initialisation (He et al., 2015), and depth invariance at initialisation by introducing a factor of $\frac{1}{\sqrt{L}}$ into the residual stream (Hayou et al., 2021; Hayou & Yang, 2023), using FLeRM to achieve invariance throughout training. We first record the function-space learning rates of the base models as in Section 4.1 using 8 random seeds, and then average over these seeds. This process is repeated for every learning rate used in our plots. When running the "scaled" models, i.e. those with altered width, depth, initialisation scale, or LoRA rank, use 40 batches of data to warm up the EMAs, and then modify the initial learning rate as specified in Algorithm (1).

We then have a choice. We can either use these learning rates for the rest of training, or we can periodically update them every 100 iterations with a single batch of data as in Algorithm (1). We found these approaches to give very similar results, so we present the results for the fixed learning rates here, and provide plots for the periodically updated

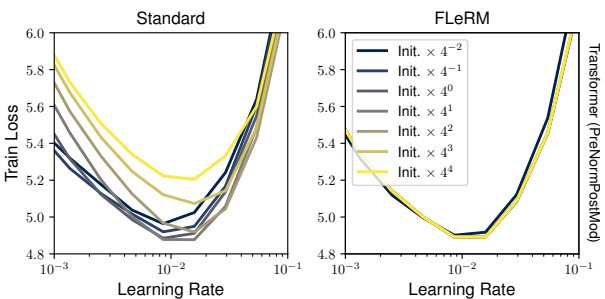

*Figure 4.* FLeRM allows us to train initialisation scale invariant networks. Top: standard practice. Bottom: FLeRM.

learning rates in Appendix C.2.

### 4.2.1. MODEL WIDTH

Figure 2 shows the effect of scaling the width on the optimal learning rate for the ResMLP and transformer models. In agreement with e.g. Yang & Hu (2022), we find that in standard practice, there is a significant shift in the optimal learning rate for all models as width increases, but when using FLeRM to normalise the layerwise parameter-space learning rates, this shift is either entirely removed or dramatically reduced. Additionally, in the transformer models, the use of FLeRM does seem to improve the loss at high widths, compared to standard practice.

### 4.2.2. MODEL DEPTH

Figure 3 shows the effect of scaling depth on the optimal learning rate for the ResMLP and transformer models. Under standard practice, the behaviour of the optimal learning rate as we increase depth varies by model. In the ResMLP, there is a significant shift in the optimal learning rate, and we observe that FLeRM brings the optimal learning rates much closer together, though for higher depths there is a slight shift towards towards larger values.

The PostNorm transformer is relatively unstable, with the loss shooting up once a certain learning rate threshold is passed. In standard practice, the location of this instability shifts to lower learning rates as depth is increased, meaning that deeper models actually have a worse optimal loss. However, with FLeRM, these instabilities all occur around the same place as the base model (either at the same learning rate or the one before it, suggesting the "true" location of the instability could be somewhere in between, at a learning rate we did not evaluate). Thus, in this setting, FLeRM gives dramatic improvements in performance for the deeper models.

In standard practice, the PreNorm transformer shows a small shift in the optimal learning rate, which is rectified by FLeRM.

The PreNormPostMod transformer is an interesting case. When scaling depth, the optimal learning rate in the standard setting is already depth-invariant, so in this setting, there is arguably no reason to apply FLeRM. Reassuringly, when we do apply FLeRM, the location of the optimal learning rate is preserved. In practice, we will usually be scaling width and depth together, so FLeRM is likely useful overall.

### 4.2.3. PARAMETER INITIALISATION SCALE

FLeRM's uses are not constrained to width and depth scaling. In the PreNormPostMod transformer, which uses residual connections of the form $x + \text{Norm}(f(x))$ and, like all 3 transformer variants, uses QK normalisation (layernorms applied after $\mathbf{W}_Q\mathbf{X}$ and $\mathbf{W}_K\mathbf{X}$), all learnable functions in the network (with the exception of the input embedding and the readout layer) are followed by a layernorm. The network should be invariant to the magnitude of these parameters, in the sense that multiplying such a weight matrix by a constant does not change the output of the network. However, as shown in Figure 4, the loss of networks trained in the standard setting varies wildly with initialisation scale, and the optimal learning rate even shifts. With FLeRM, however, we ensure that the updates are invariant in function-space, and so the loss vs. learning rate curves line up very closely for all initialisation scales, removing the need to tune this hyperparameter.

### 4.2.4. LORA ADAPTER RANK

We also investigated the use of FLeRM in a fine-tuning setting. For this, we trained autoregressively using LoRA adapters (Hu et al., 2021) on two LLMs, GPT-2 (Radford et al., 2019) and Llama-3.2-1B (Dubey et al., 2024). We experimented with ~4M token subsets of two datasets: Cold French Law (Harvard Library Innovation Lab, 2024) and Mathpile (Wang et al., 2023). For LoRA experiments, we used Adam as our base optimiser with a constant learning rate schedule, sweeping over adapter rank and the learning rate of each of the two LoRA parameters separately. We measure the base function-space learning rates for a single seed and use these with FLeRM in the scaled models. As before, we run 40 EMA warm-up iterations and then normalise the learning rates using FLeRM at the first iteration, fixing those learning rates for the rest of training. Figure 5 shows results after sweeping over the learning rate of the $B$ parameter in LoRA ($\Delta W = BA$), and in Appendix C.5 we show results for sweeping over the learning rate of $A$. See Appendix A.3 for further experimental details.

We find that in the standard setting, the optimal learning rate *increases* as we increase the LoRA rank (Figure 5); for example, the optimal learning rate for the Llama model increases by more than an order of magnitude on both datasets as we increase the LoRA rank from 2 to 32. There is also a

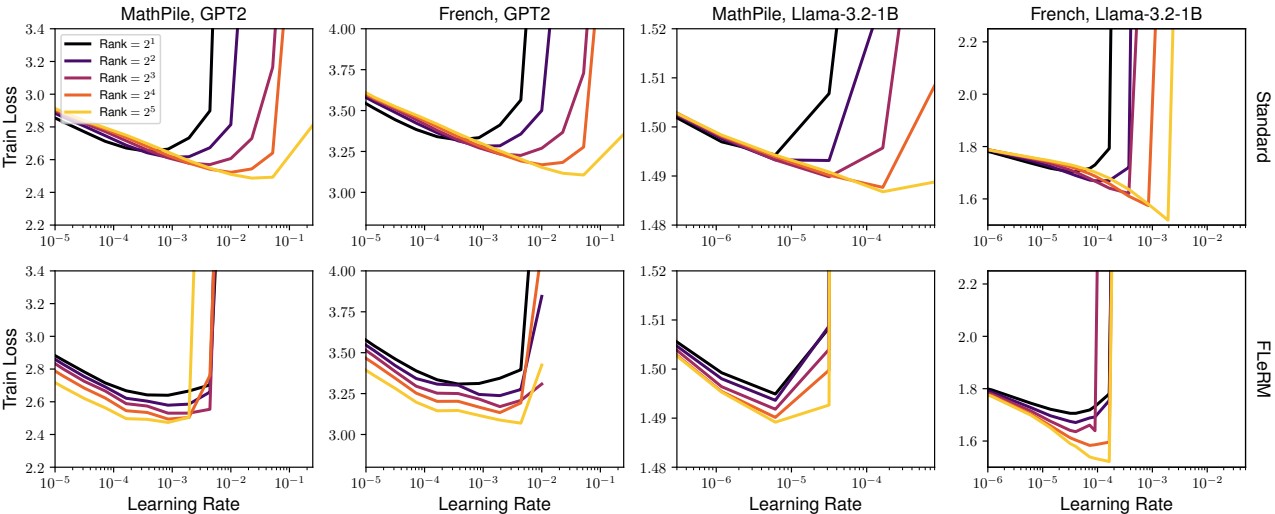

*Figure 5.* FLeRM improves optimal learning rate transfer when changing LoRA rank. The plots show the behaviour of training loss under varying the learning rate of $B$ and LoRA rank for two continual pretraining tasks. Top: standard AdamW optimiser. Bottom: (FLeRM). Some lines end abruptly for larger learning rates, indicating a numerical instability.

corresponding shift in learning rate instabilities (i.e. where the learning rate is too high). However, when using FLeRM with $r$ (Rank) equal to 2 as the "base model", the shift is either eliminated or greatly reduced, and the instabilities mostly align. Note that some data points are missing when learning rates are too high (for example for $= 2^4$ and $r = 2^5$ in the learning rate $\gtrsim 10^{-2}$ range in the bottom left plot), but this is expected because this instability is inherited from the base model.

### 4.2.5. COMPARISON TO NAÏVELY CHOSEN FUNCTION-SPACE LEARNING RATES

One might ask whether the matching the function-space learning rates of a base model is important, or if we could replace them with something simpler. To test this, we repeated the width, depth, and initialisation scale experiments from earlier, but replaced the base model's recorded function-space learning rates with uniform vectors that sum to 1. The results are shown in Figures 10, 11, and 12 (in Appendix C.3). Whilst hyperparameter transfer is still retained, the training loss is slightly worse than the equivalent experiments in Section 4.2, suggesting that the function-space learning rates induced by the Adam optimiser give benefits to performance. This ablation still shares the base function-space learning rates across new hidden blocks as depth is scaled (Section 3.3). Importantly, this means that the function-space learning rates for the embedding and readout layers are held constant even as we scale depth, because these layers are not replicated. We also ran a further ablation where all function-space learning rates are completely equal at all depths (Figure 13, Appendix C.4). In this case, the PreNormPostMod transformer no longer ex-

hibits depthwise hyperparameter transfer, which matches the findings of Large et al. (2024), who observed that the "importance" of the input and output layers must be retained as depth is scaled to achieve hyperparameter transfer. Interestingly, although hyperparameter transfer is lost, performance in the deepest PreNormPostMod model in this setting is slightly better than the standard setting, suggesting that with a more complex heuristic for matching function-space learning rates to models with new layers, it might be possible to achieve greater performance with FLeRM.

## 5. Conclusion

In this paper, we developed an efficient way to estimate the magnitude of changes in function-space caused by an update to a neural network's parameters. We used this method to analyse the dynamics of existing models, and then to modify layerwise parameter-space learning rates (FLeRM) for scaled models so that updates in function-space are scale invariant, enabling transfer of the optimal learning rate across width, depth, initialisation scale, and LoRA rank. Such a method could very useful for training very large foundation models, where scaling laws are currently derived on a case-by-case basis (e.g. see the LLama 3 technical report Dubey et al., 2024). In terms of future work, it was noted in Section 4.2.5 that a more sophisticated scheme for matching function-space learning rates to models with more layers than the base model could further enhance the performance of FLeRM. One possible approach would be to use the methods presented in this paper to study the relationship between function-space learning rates in models of various depth. However, this is beyond the scope of this work.

## Acknowledgements

Edward Milsom and Ben Anson are funded by the Engineering and Physical Sciences Research Council via the COMPASS Centre for Doctoral Training at the University of Bristol. This work was carried out using the computational facilities of the Advanced Computing Research Centre, University of Bristol - http://www.bris.ac.uk/acrc/. We would like to thank Dr. Stewart for GPU compute resources.

## Impact Statement

This paper presents work whose goal is to advance the field of Machine Learning. There are many potential societal consequences of our work, none which we feel must be specifically highlighted here.

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

# A. Model Details

### A.1. ResMLP

The base residual MLP model has 4 residual blocks, each containing a single linear layer. Every hidden layer has dimension 128, which is multiplied by the width multiplier in width-scaling experiments. In depth-scaling experiments, the number of residual blocks is multiplied by the depth multiplier. We do not use Layernorms or Batchnorms in the ResMLP (Ba et al., 2016; Ioffe & Szegedy, 2015).

We initialise all weight matrices using Kaiming / He initialisation (He et al., 2015), that is, IID Gaussian weights with $\frac{1}{\texttt{fan\_in}}$ variance, multiplied by an activation-function-specific scalar (2 in the case of ReLU, 1 for no activation). Biases are initialised to 0.

To ensure depth invariance at initialisation, we introduce a factor of $\frac{1}{\sqrt{L}}$ into each layer's weight matrix. At initialisation, this is equivalent to multiplying by $\frac{1}{\sqrt{L}}$ in the forward pass as in Bordelon et al. (2023), but does not continue to affect the forward passes during training, allowing us to better isolate the effect of FLeRM.

We optimise the model using Adam with default settings (other than the learning rate which we set using the methods detailed in this paper). We train for 50 epochs on the CIFAR-10 dataset (Krizhevsky & Hinton, 2009).

### A.2. Transformer

The base transformer model is a decoder-only transformer with 2 self-attention + feedforward layers, query-key normalisation (Dehghani et al., 2023, applying layernorm to the queries and keys before using them in multihead attention), and layernorms with affine transformations disabled. We compare 3 different types of layernorm in our experiments: PostNorm is $\text{Norm}(x + f(x))$, PreNorm is $x + f(\text{Norm}(x))$, and PreNormPostMod is $x + \text{Norm}(f(x))$.

When scaling width, we multiply the number of heads, the embedding dimension $d_{\text{model}}$, and the feedforward hidden dimension $d_{\text{ff}}$ by the width multiplier. The base model has $d_{\text{model}} = 128$, $d_{\text{ff}} = 512$, and 2 attention heads per layer. When scaling depth, we multiply the number of transformer "blocks", consisting of self-attention and a feedforward network, as detailed in Vaswani et al. (2017), by the depth multiplier.

We initialise all weight matrices using Kaiming / He initialisation (He et al., 2015), that is, IID Gaussian weights with $\frac{1}{\texttt{fan\_in}}$ variance, multiplied by an activation-function-specific scalar (2 in the case of ReLU, 1 for no activation). Biases are initialised to 0.

To ensure the model is invariant to depth at initialisation, we multiply the residual branch by $\frac{1}{\sqrt{L}}$ during the forward pass, as in Bordelon et al. (2023). More specifically, PostNorm becomes $\text{Norm}(x + \frac{1}{\sqrt{L}}f(x))$, PreNorm becomes $x + \frac{1}{\sqrt{L}}f(\text{Norm}(x))$, and PreNormPostMod becomes $x + \frac{1}{\sqrt{L}}\text{Norm}(f(x))$. Note that with PostNorm and PreNorm we could have absorbed this factor into the weights of the module $f$ and therefore avoided altering the forward pass computation, because $f$ is at the end of the residual stream. However, in PreNormPostMod, the layernorm is at the end of the residual stream, so its initialisation will always have unit size no matter how we initialise $f$, therefore requiring us to use the factor $\frac{1}{\sqrt{L}}$ during the forward pass. We therefore decided to treat all transformer variants the same in this regard.

We train the transformer on roughly $\frac{1}{10}$ of the Wikitext-103 dataset (Merity et al., 2016), using a batch size of 20 and a sequence length of 256. We tokenise the dataset using the GPT2 tokeniser from the HuggingFace transformer library (Wolf et al., 2020). We optimise the model using Adam with default settings (other than the learning rate which we set using the methods detailed in this paper). We train for 1 epoch (i.e. we only observe each token once).

### A.3. LoRA adapters

For the LoRA experiments in Section 4.2.4, we trained LoRA adapters on continual pretraining tasks. The LoRA adapters were initialized using the `gaussian` initialization provided by Hugging Face's `peft` library (Mangrulkar et al., 2022). We experimented with two models, GPT2 and Llama-3.2-1B. We added LoRA adapters to the default modules of GPT2, and the `q/v/k/o_proj` modules of Llama-3.2-1B. We trained for 500 iterations with a batchsize of 8 and sequence length 512. The datasets used were ∼4M token subsets of Cold French Law (Harvard Library Innovation Lab, 2024) and Mathpile (Wang et al., 2023).

A LoRA adapter is formed of two parameters $B$ and $A$, with $\Delta W = BA$. When sweeping learning rates, we vary the learning rates of each parameter ($A/B$) separately, while keeping the learning rate of the other parameter fixed. When sweeping for $B$, we used a fixed learning rate of $10^{-4}$ for $A$. When sweeping for $A$, we fixed the learning rate of $B$ as follows: $10^{-3}$ for GPT-2, $10^{-4}$ for Llama-3.2-1B / Cold French Law, and $5 \times 10^{-5}$ for Llama-3.2-1B/Mathpile .

As in the transformer and ResMLP experiments, we wish to warm-up the EMAs of FLeRM with 40 batches of data, and then use FLeRM to normalise the learning rates at the initial iteration, fixing these learning rates for the rest of training. However, we cannot immediately use our method FLeRM when using LoRA adapters. This is due to the way LoRA adapters (following the original work of Hu et al. (2021)) are initialized. Since the parameter $B$ is initialized to zero (to ensure that $\Delta W = 0$ at the beginning of training), the gradient of $A$ is 0 for the first iteration. This is problematic because it implies $\|\Delta_\ell f\|_{\text{RMS}} = 0$, leading to a division by zero in Algorithm 1. As a workaround, we warm-up FLeRM on the fifth iteration, and record / set the learning rates on the 6th iteration. We find that in practice, this still gives adequate hyperparameter transfer. In the LoRA experiments, we measure the base function-space learning rates for a single seed and use these with FLeRM in the scaled models. This is because the base model is a lot more expensive than in the other experimental setups. Only using one seed for the base function-space learning rates did not appear to hinder FLeRM's ability to enabled LoRA-rank transfer.

## B. Kronecker factored covariance approximation for tensor-valued parameters

Previous work has extended the matrix-normal distribution to tensors, with one covariance matrix per dimension (e.g. Manceur & Dutilleul, 2013; Hoff, 2010). We can use this to extend the methods in Section 3.2 to tensor-valued parameters.

Suppose we have a $D$-dimensional tensor $\mathbf{X} \in \mathbb{R}^{n_1 \times \cdots \times n_D}$, where we will use $X_{i_1, \ldots, i_D}$ to denote the $(i_1, \ldots, i_D)^{\text{th}}$ element. Extend the $\text{vec}(\cdot)$ operation to tensors in the obvious way, i.e. form a vector by taking elements $(1, 1, \ldots, 1, 1)$ to $(n_1, 1, \ldots, 1, 1)$, then $(1, 2, \ldots, 1, 1)$ to $(n_1, 2, \ldots, 1, 1)$ etc. systematically iterating along the dimensions until you reach the final element $(n_1, \ldots, n_D)$. We say the vector $\mathbf{X}$ is **tensor-normal distributed** if

$$\text{vec}(\mathbf{X}) \sim \mathcal{N}(\text{vec}(\mathbf{M}), \mathbf{U}^{(D)} \otimes \cdots \otimes \mathbf{U}^{(1)}) \tag{20}$$

where $\mathbf{M} \in \mathbb{R}^{n_1, \ldots, n_D}$ is the mean tensor, and $\mathbf{U}^{(1)} \in \mathbb{R}^{n_1 \times n_1}, \ldots, \mathbf{U}^{(D)} \in \mathbb{R}^{n_D \times n_D}$ are covariance matrices for the $D$ different dimensions. In other words, assuming $\mathbf{M} = 0$, the second-order moments / covariance of any two elements in the tensor factorises across dimensions:

$$\mathbb{E}(X_{i_1, \ldots, i_D} X_{j_1, \ldots, j_D}) = \prod_{d=1}^{D} U^{(d)}_{i_d, j_d}. \tag{21}$$

For matrix-normal distributions, we had identities for the second order matrix products $\mathbb{E}[\mathbf{X}\mathbf{X}^T] = \mathbf{U}\text{tr}(\mathbf{V})$ and $\mathbb{E}[\mathbf{X}^T\mathbf{X}] = \mathbf{V}\text{tr}(\mathbf{U})$, which we used in Section 3.2 to figure out how to estimate the covariance matrices (or more specifically, the sum over all pairs of elements). Do similar identities hold for the tensor normal distribution? It turns out yes (e.g. Proposition 2.1 of Hoff, 2010).

Consider the following shorthand for "contracting" two tensors over all but one of their dimensions $d$, kind of like a generalisation of matrix multiplication

$$(\mathbf{A} \times_{\text{d}} \mathbf{B})_{ij} = \sum_{k_1 \ldots k_{d-1} k_{d+1} \ldots k_D} A_{k_1 \ldots k_{d-1} i k_{d+1} \ldots k_D} B_{k_1 \ldots k_{d-1} j k_{d+1} \ldots k_D} \tag{22}$$

i.e. assuming $\mathbf{A}$ and $\mathbf{B}$ have conformable shapes, you take "dot products" over all dimensions except $d$. Then for a

tensor-normal distributed $\mathbf{X}$ we have the "second-order moments"

$$\mathbb{E}(\mathbf{X} \times_d \mathbf{X})_{ij} = \sum_{k_1 \ldots k_{d-1} k_{d+1} \ldots k_D} \mathbb{E}(X_{k_1 \ldots k_{d-1} i k_{d+1} \ldots k_D} X_{k_1 \ldots k_{d-1} j k_{d+1} \ldots k_D}) \tag{23a}$$

$$= \sum_{k_1 \ldots k_{d-1} k_{d+1} \ldots k_D} U_{ij}^{(d)} \prod_{d' \neq d} U_{k_{d'}, k_{d'}}^{(d')} \tag{23b}$$

$$= U_{ij}^{(d)} \prod_{d' \neq d} \left( \sum_{k_{d'}} U_{k_{d'}, k_{d'}}^{(d')} \right) \tag{23c}$$

$$= U_{ij}^{(d)} \prod_{d' \neq d} \mathrm{tr}(\mathbf{U}^{(d')}) \tag{23d}$$

and so we have

$$\mathbb{E}(\mathbf{X} \times_d \mathbf{X}) = \mathbf{U}^{(d)} \prod_{d' \neq d} \mathrm{tr}(\mathbf{U}^{(d')}). \tag{24}$$

As before, we want to compute the sum over all elements of the covariance matrix. The covariance matrix for the vectorised tensor is $\mathbf{U}^{(D)} \otimes \cdots \otimes \mathbf{U}^{(1)}$, which means it contains precisely all the possible products of D elements, one from each $\mathbf{U}^{(d)}$. Hence we wish to compute

$$\prod_{d=1}^{D} \left( \sum_{i_d, j_d} U_{i_d j_d}^{(d)} \right) = \prod_{d=1}^{D} \frac{\sum_{i_d, j_d} \mathbb{E}(\mathbf{X} \times_d \mathbf{X})_{i_d j_d}}{\prod_{d' \neq d} \mathrm{tr}(\mathbf{U}^{(d')})} \tag{25a}$$

$$= \frac{\prod_{d=1}^{D} \sum_{i_d, j_d} \mathbb{E}(\mathbf{X} \times_d \mathbf{X})_{i_d j_d}}{\prod_{d=1}^{D} \mathrm{tr}(\mathbf{U}^{(d)})^{D-1}}. \tag{25b}$$

As before, we can express the denominator in terms of $\mathbf{X}$. Note that

$$\prod_{d=1}^{D} \mathrm{tr}(\mathbf{U}^{(d)}) = \mathrm{tr} \left( \mathbf{U}^{(1)} \prod_{d=2}^{D} \mathrm{tr}(\mathbf{U}^{(d)}) \right) \tag{26a}$$

$$= \mathrm{tr} \left( \mathbb{E}(\mathbf{X} \times_1 \mathbf{X}) \right) \tag{26b}$$

$$= \sum_{i} \mathbb{E}(\mathbf{X} \times_1 \mathbf{X})_{ii} \tag{26c}$$

$$= \sum_{i} \sum_{k_2 \ldots k_D} \mathbb{E}(X_{i k_2 \ldots k_D} X_{i k_2 \ldots k_D}) \tag{26d}$$

$$= \mathbb{E} \sum_{k_1 \ldots k_D} X_{k_1 \ldots k_D}^2 \tag{26e}$$

$$= \mathbb{E} ||\mathbf{X}||_{\mathcal{F}}^2 \tag{26f}$$

where we have used the Frobenius norm to represent the sum of all squared elements of the tensor. So the denominator will be $\mathbb{E}(||\mathbf{X}||_{\mathcal{F}}^2)^{D-1}$.

Also similar to the matrix-normal case, the numerator is very cheap to compute. Observe that

$$\sum_{i_d, j_d} \mathbb{E}(\mathbf{X} \times_d \mathbf{X})_{i_d j_d} = \mathbb{E} \sum_{i_d, j_d} \sum_{k_1 \ldots k_{d-1} k_{d+1} \ldots k_D} X_{k_1 \ldots k_{d-1} i_d k_{d+1} \ldots k_D} X_{k_1 \ldots k_{d-1} j_d k_{d+1} \ldots k_D} \tag{27a}$$

$$= \mathbb{E} \sum_{k_1 \ldots k_{d-1} k_{d+1} \ldots k_D} \sum_{i_d, j_d} X_{k_1 \ldots k_{d-1} i_d k_{d+1} \ldots k_D} X_{k_1 \ldots k_{d-1} j_d k_{d+1} \ldots k_D} \tag{27b}$$

$$= \mathbb{E} \sum_{k_1 \ldots k_{d-1} k_{d+1} \ldots k_D} \left( \sum_{i_d} X_{k_1 \ldots k_{d-1} i_d k_{d+1} \ldots k_D} \right)^2 \tag{27c}$$

which in pseudocode can be written as $\mathbb{E}($X.sum(d).square().sum()$)$.

Hence, using pseudocode to make things clearer, our normaliser can be estimated as

$$||\Delta_\ell \mathbf{f}||_{\text{RMS}} \approx \sqrt{\frac{\prod_{d=1}^{D} \mathbb{E}(\texttt{Z.sum(d).square().sum()})}{\mathbb{E}(\texttt{Z.square().sum()})^{D-1}}} \tag{28}$$

where Z is our tensor of updates times phi-gradients, and the expectation symbols $\mathbb{E}$ tell us what we're taking EMAs of. Remarkably, we can see that we only have to track $D+1$ scalar EMAs, and updating these EMAs only requires simple sum() and square() operations. For numerical stability, it is wise to compute this quotient in log-domain.

## C. Extra Plots

### C.1. Full function-space learning rate vs time plot

Figure 6 shows the function-space learning rates over time as in Figure 1 in the main text, but also with the Transformer (PreNorm) and Transformer (PreNormPostMod) models. The plots for the three transformer layernorm variants all show very similar behaviour.

### C.2. FLeRM with periodic updates to the learning rate

In the FLeRM experiments in the main text (Section 4.2) we only used FLeRM to modify the learning rate during the first step of training, and then used that learning rate for the rest of training. There is nothing to stop us from periodically updating the learning rate throughout training using FLeRM if necessary, e.g. if we think the dynamics of training might be affected by width or depth in a time-dependent manner. Here we present plots similar to Figures 2, 3, and 4 from the main text, but in addition to using FLeRM to compute the learning rate at the start, we also update the learning rate every 100 iterations using a single batch of data and an EMA decay rate $\beta = 0.9$. The resultant width, depth, and initialisation scale transfer plots are given in Figures 7, 8, and 9. The plots are very similar to those using the static learning rate, suggesting that the effects of increasing width, depth, or initialisation scale can be effectively cancelled-out using a learning rate computed purely at the start of training. This agrees with prior work like µP (Yang & Hu, 2022) and Module (Large et al., 2024), which both achieve hyperparameter transfer in width and depth scaling settings using static learning rates.

### C.3. FLeRM with equal base model function-space learning rates (but still splitting them in the same way as depth scales

Here we replace the recorded base model function-space learning rates with equal values for each layer, that add up to 1. As depth is increased, with still split the function-space learning rates up between the new hidden blocks (described in Section 3.3) as we did in the main experiments. See Figures (10), (11), and (12).

### C.4. FLeRM with equal base function-space learning rates that are ALWAYS equally divided across layers, even with scaled depth

This ablation is similar to the equal base function-space learning rates ablation in Appendix C.3, but instead of sharing the values across the new hidden layer blocks as described in Section 3.3, we simply set all the "base" function-space learning rates for the deeper model to be equal and sum to 1. Note that a key difference in this approach is that the input embedding and readout layer's base function-space learning rate is going to shrink as the model gets deeper, whereas in the ablation in Appendix C.3, the base function-space learning rates for the input embedding and readout layer remained constant, whilst only the hidden layers got diluted when scaling depth. As suggested in previous work (Large et al., 2024), ensuring input embeddings and readout layers retain their "importance" during training in deeper models is critical for achieving depth transfer, and we observe in this ablation (Figure 13, only depth scaling is changed from the other equal function-space learning rate ablation) that we no longer have hyperparameter transfer across depth. Interestingly, however, the performance of the deepest PreNormPostMod model is better in this ablation than in the other FLeRM experiments, actually marginally beating the standard practice model too, suggesting further refine of the scheme for splitting base FSLRs as depth increases could be beneficial. However, performance in this ablation is also far worse on other models like PostNorm at high depths.

## C.5. LoRA

In Section 4.2.4, we showed results for sweeping over the learning rate of the LoRA adapters' B parameter, with a fixed learning rate for A. Here, we show results for the reverse: sweeping over the learning rate of A with a fixed learning rate for B. Figure 14 shows the results. Interestingly, we achieve learning rate transfer with the Standard AdamW optimiser, mainly because varying the learning rate of $A$ does not change the performance of the model (until it becomes too large). This suggests that selecting the learning rate of $A$ is not very important compared to selecting the learning rate of $B$. We find that with FLeRM, we mostly preserve the learning rate transfer, and we also observe an instability shift to the left.

## C.6. Bias and variance of estimator under difference covariance assumptions

In Figure 15 we show that our proposed Kronecker-factored assumption in Section 3.2 greatly reduces the variance of the function-space learning rate estimator whilst avoiding the large bias an oversimplifying assumption such as an IID covariance structure would introduce. Here the "no assumption" estimator is used as the "true" value for computing the biases, as it is an unbiased estimator.

## C.7. Test loss

In Figure 16 we show the width transfer plot from the main text but with test losses instead of train losses. As expected, the test loss plots for the Transformer models look very similar to the main text, since we are only training for 1 epoch and therefore expect test loss to look like train loss. However, the ResMLP is trained for multiple epochs on CIFAR-10, and so complex overfitting patterns occur, which probably explains why the test loss curves do not line up perfectly, though the optima are closer together with FLeRM than with standard practice.

## C.8. Cosine Annealing Scheduler

For simplicity we used a constant LR scheduler in the main text. To verify that our method works with LR schedulers, we reran the width transfer experiments using a cosine annealing scheduler. Since FLeRM modifies the layerwise learning rates, we record the ratio of the current scheduled learning rate with the starting learning rate in the base model, then apply these ratios as the scheduler in the scaled model. This is equivalent to scheduling the "target" function-space learning rate for each layer. The results are shown in Figure 17.

## C.9. Elementwise Affine Transformations

For simplicity in prototyping, elementwise affine transformations in layernorms were disabled in the main experiments. To verify our method still works with these enabled, we reran the width transfer experiment for the PreNormPostMod transformer, and, as seen in Figure 18, found no issues.

## C.10. Simultaneous Width + Depth Scaling

To test whether our method still works when simultaneously scaling width and depth, we ran the PreNormPostMod transformer experiment again, scaling the width+depth, and found that hyperparameter transfer still holds in this setting. The results are shown in Figure 19

## C.11. Other optimisers

To test whether our method still works when using optimisers with Adam, we repeated the Transformer (PreNormPostMod) width transfer experiments for a range of different optimisers.

In Figure 20 we used SGD with momentum. In Figure 21 we used SignSGD. In Figure 22 we used AdamW. In Figure 23, we used Adamax. In Figure 24, we used Adagrad. We can see that FLeRM improves hyperparameter transfer in all cases. In SGD, the curves are very noisy both with and without FLeRM, matching the common practice of not using SGD for transformers. However, even though the curves are less neat than the other optimisers, we still see that FLeRM aligns the curves better.

To demonstrate this is a problem with using SGD with transformers and not FLeRM, in Figure 25 we ran the ResMLP width transfer experiment with SGD + momentum, where we observely that FLeRM very cleanly improves hyperparameter

transfer.

## D. Exploiting known structure for lower variance function-space learning rate estimation

A typical neural network ends in a "readout" layer $\{\mathbf{W}^L, \mathbf{b}^L\}$, mapping from the hidden dimension $d_{\text{model}}$ to the number of output classes / vocab size $K$ via the transformation

$$\mathbf{f}_n = \mathbf{W}^L \mathbf{h}_n^{L-1} + \mathbf{b}^L \tag{29}$$

where $\mathbf{h}_n^{L-1}$ is the output of the previous layer for the $n^{\text{th}}$ datapoint.

This layer can be very large if $K$ is large (the vocab size of an LLM), and can have large effects on the training dynamics of the neural network (Fig. 1). It is therefore important to ensure its function-space learning rate estimate is as accurate as possible. Luckily, it turns out that by nature of being the final layer, there is extra independence structure in the function-space learning rate estimation procedure we can exploit to lower the variance.

Subbing Eq. 29 into Eq. 3, we have

$$\phi = \frac{1}{\sqrt{NK}} \sum_{nk} \omega_{nk} \left( b_k + \sum_{\alpha} W_{k\alpha}^L h_{n\alpha}^{L-1} \right) \tag{30}$$

and subsequently subbing Eq. 3 into Eq. 8, considering the function-space learning rate for the readout weight matrix $\mathbf{W}^L \in \mathbb{R}^{K \times d_{\text{model}}}$, we obtain

$$Z_{ij}^{(\mathbf{W}^L)} = \Delta W_{ij}^L \frac{1}{\sqrt{NK}} \sum_n \omega_{ni} h_{ni}^{L-1}. \tag{31}$$

Note that this only includes a single random variable $\omega_{ni}$. This means the random variables $\{Z_{ij}^{(\mathbf{W}^L)}\}_{ij}$ are independent for different values of $i$ (i.e. between different rows), but dependent for different values for $j$ (i.e. between different columns). Hence we can assume independence between the rows of $Z_{ij}^{(\mathbf{W}^L)}$, drastically reducing the variance of our estimator.

In particular, this corresponds to assuming that $\mathbf{U}$ is diagonal in Section 3.2, which results in $\|\Delta_\ell \mathbf{f}\|_{\text{RMS}}^2 = \left( \sum_i U_{ii} \right) \left( \sum_{jj'} V_{jj'} \right) = \text{tr}(\mathbf{U}) \left( \sum_{jj'} V_{jj'} \right)$ and therefore our estimate becomes

$$\|\Delta_{\mathbf{W}^L} \mathbf{f}\|_{\text{RMS}}^2 = \text{tr}(\mathbf{U}) \frac{\mathbb{E}\left( \sum_{jj'} [\mathbf{Z}^T \mathbf{Z}]_{jj'} \right)}{\text{tr}(\mathbf{U})} = \mathbb{E}\left( \sum_{jj'} [\mathbf{Z}^T \mathbf{Z}]_{jj'} \right). \tag{32}$$

Similarly for the biases $\mathbf{b}^L \in \mathbb{R}^K$ we have

$$Z_i^{(\mathbf{b}^L)} = \Delta b_i^L \frac{1}{\sqrt{NK}} \sum_n \omega_{ni} \tag{33}$$

and so $Z_i^{(\mathbf{b}^L)}$ are independent for different $i$, and so we can assume the covariance matrix is diagonal. If we assume a diagonal covariance structure over $Z_i^{(\mathbf{b}^L)}$, i.e. $\boldsymbol{\Sigma} := \text{diag}(\boldsymbol{\sigma}^2)$, then

$$\|\Delta_\ell \mathbf{f}\|_{\text{RMS}}^2 = \sum_{ii'} \Sigma_{ii'} = \sum_{i=1}^n \sigma_i^2 = \sum_{i=1}^K \mathbb{E}\left( z_i^{(\mathbf{b}^L)} \right)^2 = \mathbb{E}\left[ \sum_{i=1}^K \left( z_i^{(\mathbf{b}^L)} \right)^2 \right]. \tag{34}$$

One might ask at this point how this differs from an IID assumption, i.e. $\boldsymbol{\Sigma} := \sigma^2 \mathbf{I}$. With an IID assumption, we would have

$$\|\Delta_\ell \mathbf{f}\|_{\text{RMS}}^2 = \sum_{ii'} \Sigma_{ii'} = n\sigma^2 = n\mathbb{E}\left( z_1^{(\mathbf{b}^L)} \right)^2 = n\mathbb{E}\left[ \frac{1}{n} \sum_{i=1}^K \left( z_i^{(\mathbf{b}^L)} \right)^2 \right] = \mathbb{E}\left[ \sum_{i=1}^K \left( z_i^{(\mathbf{b}^L)} \right)^2 \right] \tag{35}$$

which is exactly the same! It turns out that, because we only care about estimating the sum over the variances, assuming that $Z_i^{(\mathbf{b}^L)}$ are IID or just independent are equivalent. Intuitively, consider the difference between making "no assumption" on the covariance matrix versus a diagonal assumption, if we care about the sum over all covariance elements. The diagonal assumption allows us to remove cross-terms from our estimate, since we know their true values are all zero, and hence we remove the variance from the randomness in those cross-terms, giving us a lower variance estimator. But if we consider the remaining positive sum over the variance (diagonal) terms, knowing they're all equal (an IID assumption) doesn't provide us with any more useful information, because it doesn't tell us any more about the value of the total sum, and it doesn't tell us any more about which values we should pay more attention to (in fact, it tells us to treat them all equally, which is what we already did when we only knew that the covariance was diagonal). If we had more specific information, such as "the last element's variance forms 90% of the total variance" then we could use more sophisticated weighted averages to get a lower variance estimate.

In Figure 26, we repeat the width transfer experiments from the main paper but using the tricks we just described, and find that some of the instability / noise in the ResMLP FLeRM plot is now gone.

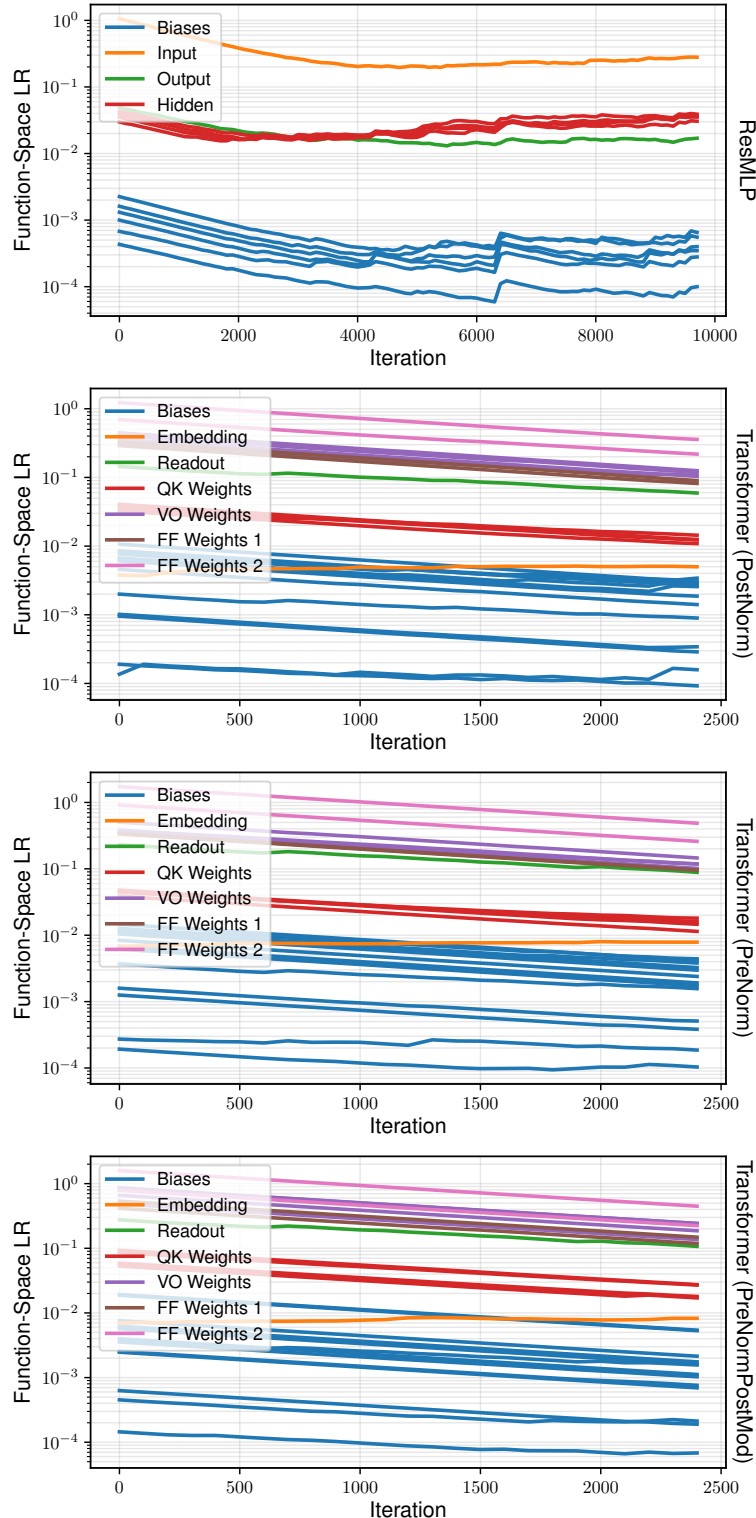

*Figure 6.* Function-space learning rates over time, measured using our approach, for the ResMLP model (row 1) and the transformer models with different layernorm strategies (rows 2, 3, and 4).

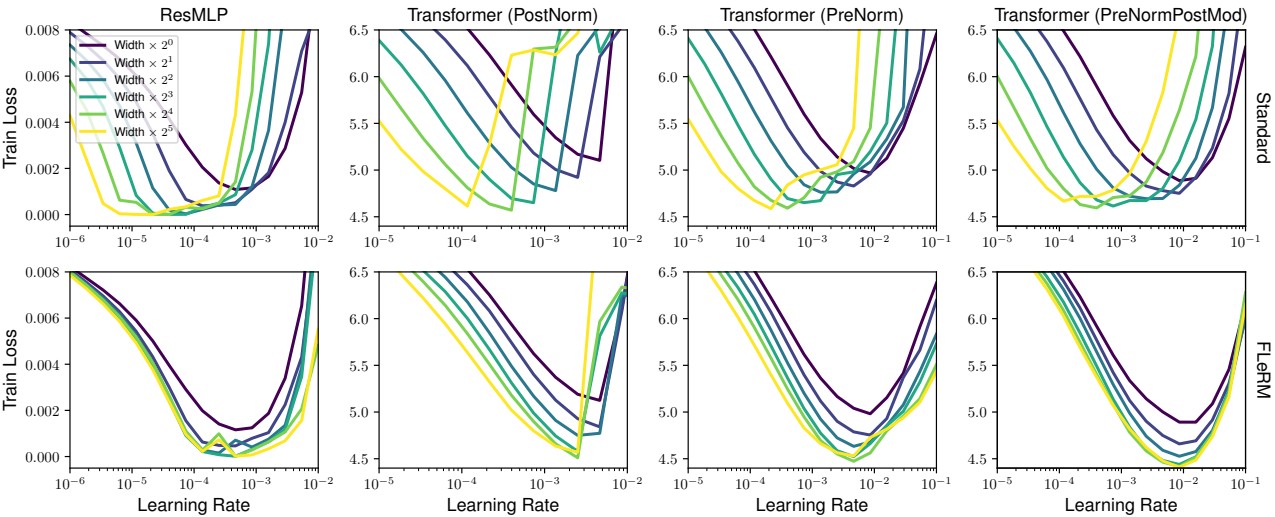

*Figure 7.* Normalising function-space learning rates periodically using FLeRM gives very similar width transfer to normalising using FLeRM at the first iteration only (see Figure 2). Top: standard practice. Bottom: our scheme (FLeRM). PreNorm, PostNorm, and PreNormPostMod are different Layernorm configurations described in Section A

.

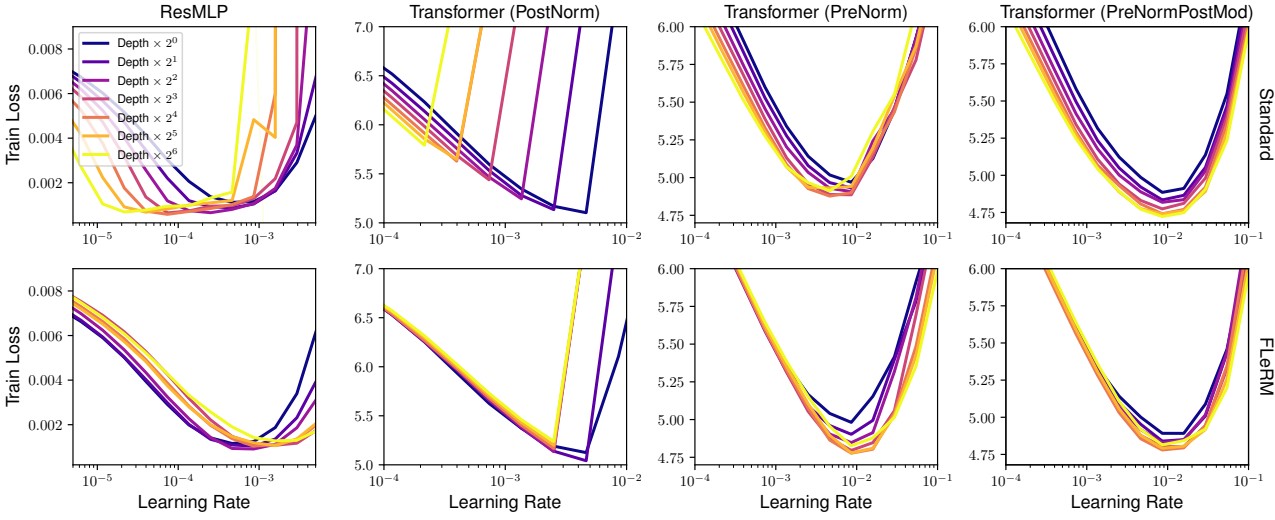

*Figure 8.* Normalising function-space learning rates periodically using FLeRM gives very similar depth transfer to normalising using FLeRM at the first iteration only (see Figure 3). Top: standard practice. Bottom: our scheme (FLeRM).

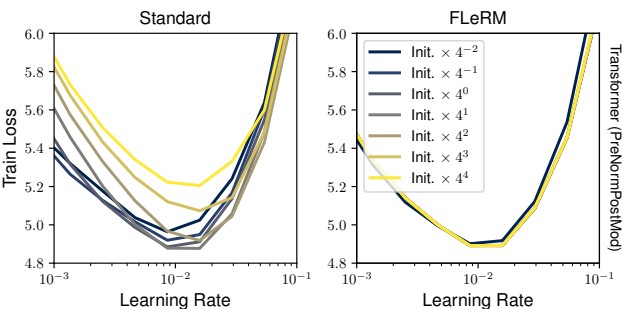

*Figure 9.* Normalising function-space learning rates periodically using FLeRM gives very similar initialisation scale transfer to normalising using FLeRM at the first iteration only (see Figure 4). Top: standard practice. Bottom: our scheme (FLeRM).

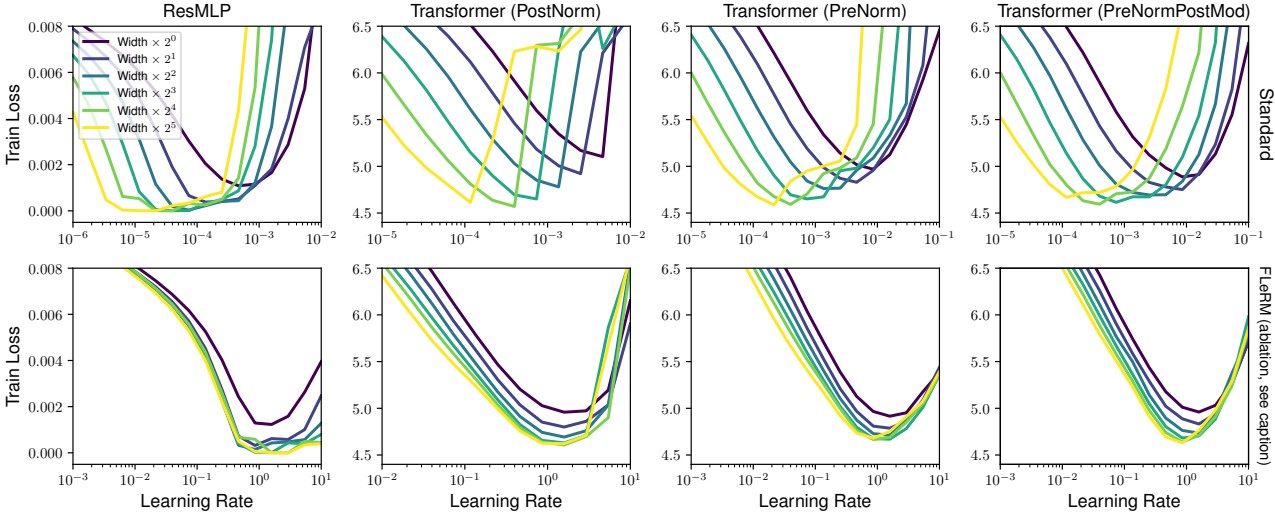

*Figure 10.* Equal base FSLR ablation (width): Using equal base model function-space learning rates results in a degradation in performance in some of the models, compared to the main experiments. Top: standard practice. Bottom: our scheme (FLeRM) with equal base model function-space learning rates.

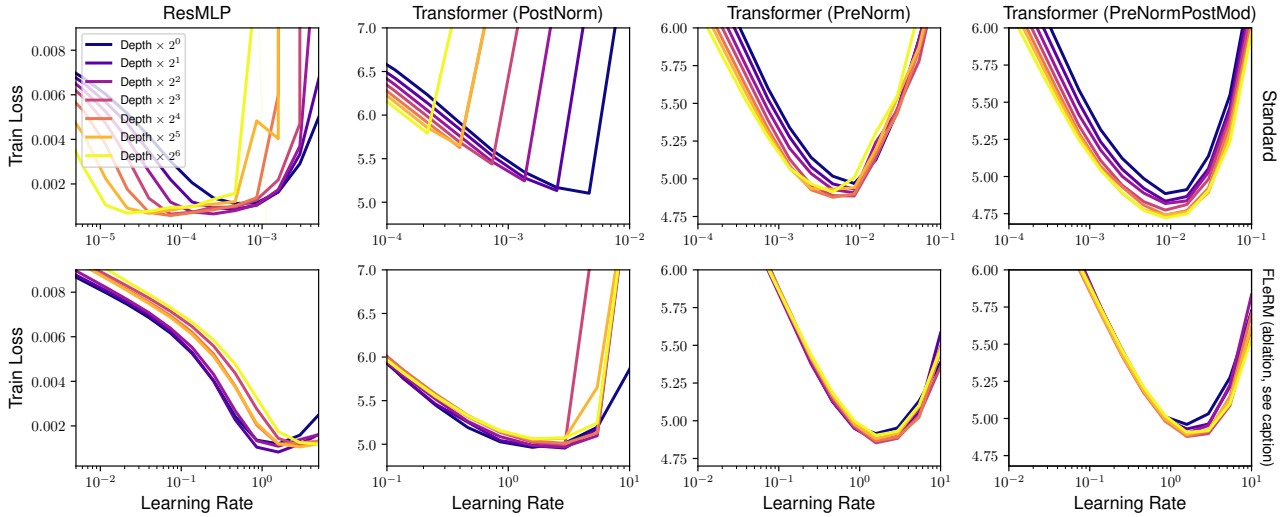

*Figure 11.* Equal base FSLR ablation (depth): Using equal base model function-space learning rates results in a degradation in performance in some of the models, compared to the main experiments. Top: standard practice. Bottom: our scheme (FLeRM) with equal base model function-space learning rates.

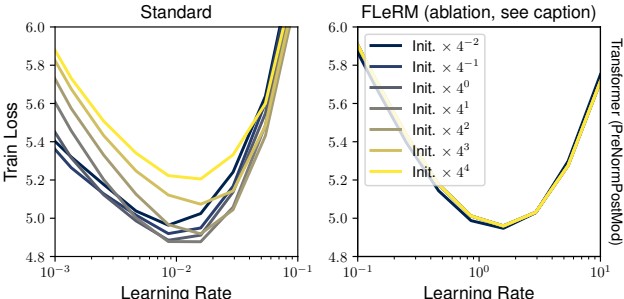

*Figure 12.* Equal base FSLR ablation (init. scale): Using equal base model function-space learning rates results in a degradation in performance in some of the models, compared to the main experiments. Top: standard practice. Bottom: our scheme (FLeRM) with equal base model function-space learning rates.

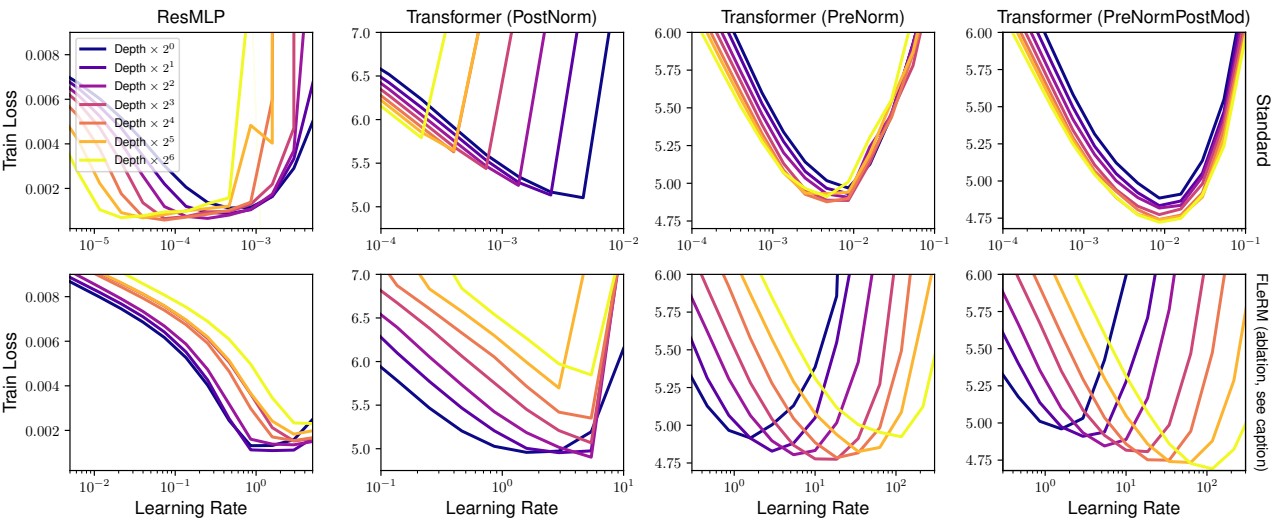

*Figure 13.* Equal base FSLR ablation where we don't carefully retain the importance of the input embedding and readout layers (depth): Just setting the base model FSLRs to be equal and sum to one, no matter what the depth is, means that we no longer have depthwise hyperparameter transfer in the PreNormPostMod model. Top: standard practice. Bottom: our scheme (FLeRM) with equal base model function-space learning rates that don't try to account for depth increase.

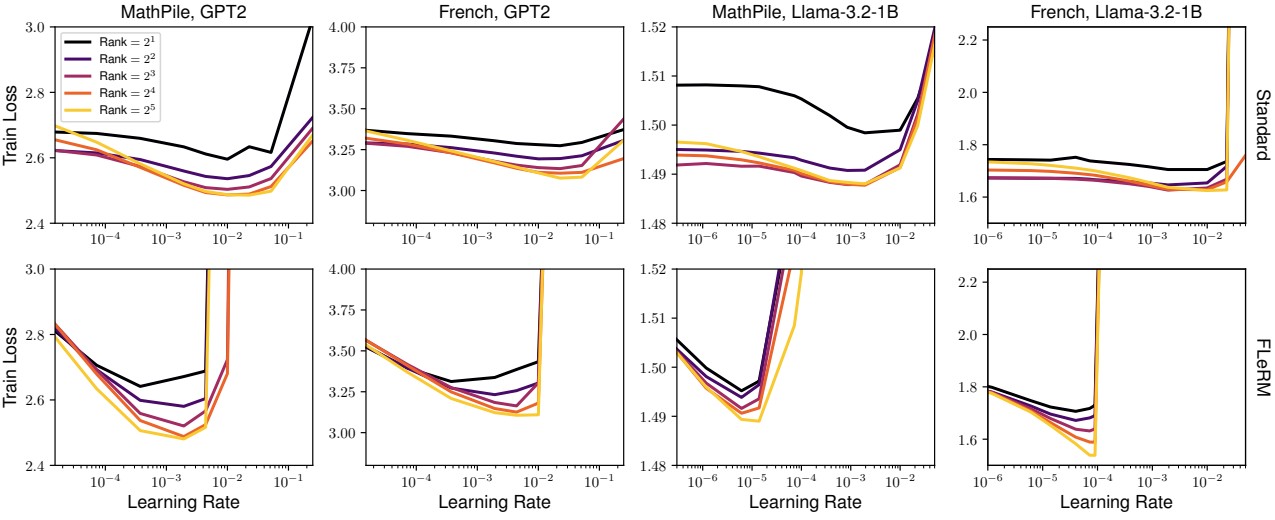

*Figure 14.* Behaviour of training log likelihood loss under varying the learning rate of $A$ and LoRA rank for two continual pretraining tasks. The top row shows results from a standard setup with AdamW, the bottom row shows our method, FLeRM.

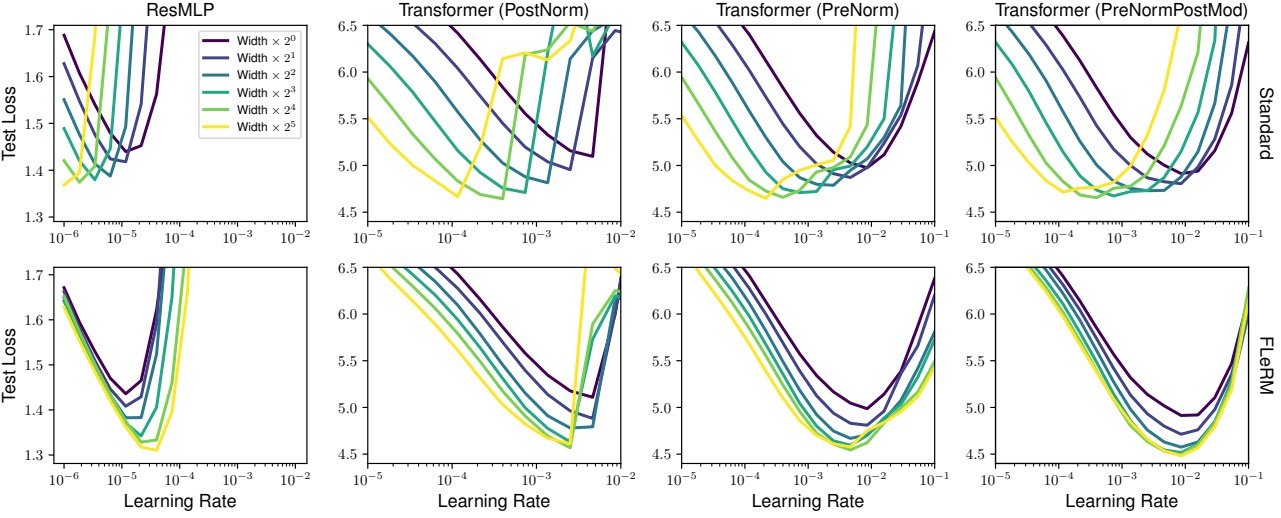

Layer layers.0.ff.net.0.weight

*Figure 15.* Comparison of bias and variance of the function-space learning rate estimator when using different covariance matrix / dependence assumptions in Section 3.1 / 3.2. Computed over 10,000 batches to estimate the FSLRs at the first step of training for a feedforward layer in the network. Bias is taken as the absolute difference from "No assumption", since no assumption (i.e. ignoring Section 3.2 and estimating the quantity directly) is an unbiased estimator. We see that an IID assumption has extremely low variance (not even visible on the plot), but very high bias, whilst the unbiased "no assumption" estimator has very high variance. The KFAC estimator proposed in Section 3.2 has a small amount of bias and much smaller variance than the "no assumption" estimator.

*Figure 16.* Width transfer plot for plotting **test loss** instead of train loss.

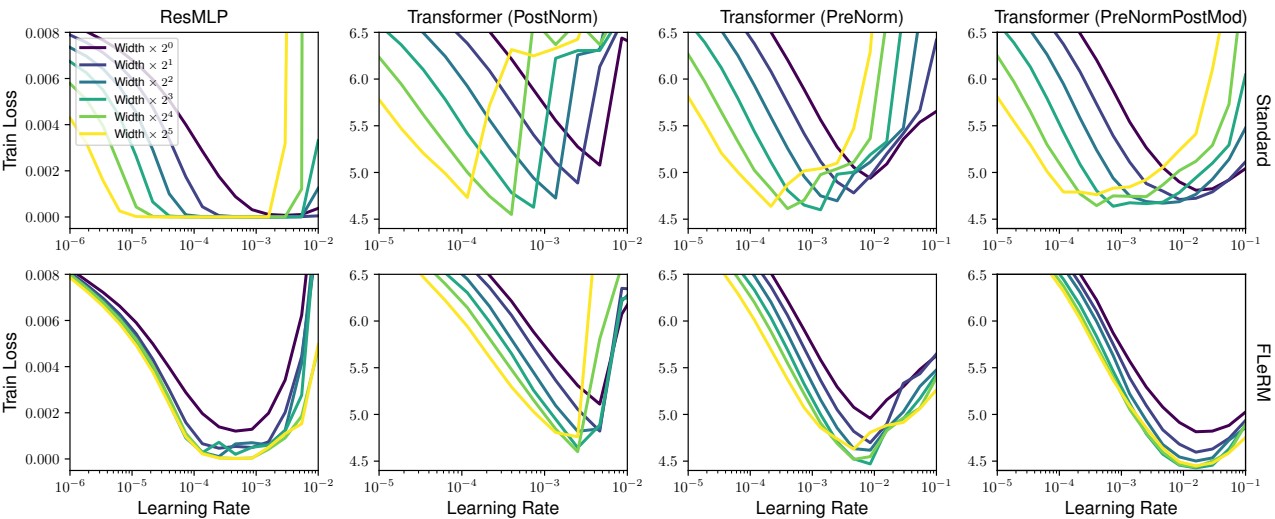

*Figure 17.* Width transfer plot for Transformer (PreNormPostMod) using a **CosineAnnealingLR scheduler**.

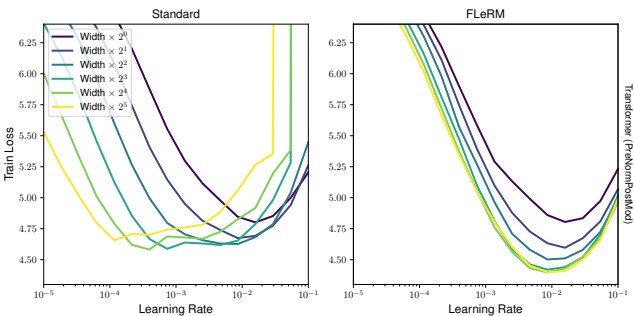

*Figure 18.* Width transfer plot for Transformer (PreNormPostMod) with **elementwise affine transformations enabled** in the Layernorms.

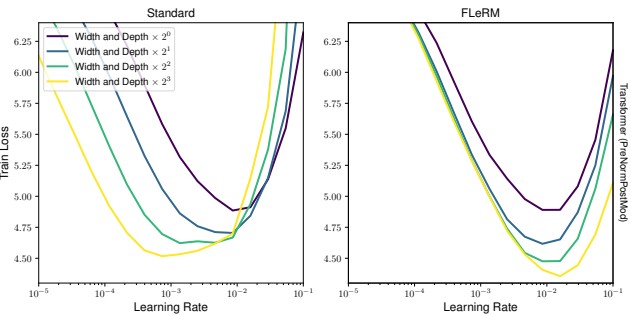

*Figure 19.* **Width + Depth simultaneous scaling** transfer plot for Transformer (PreNormPostMod).

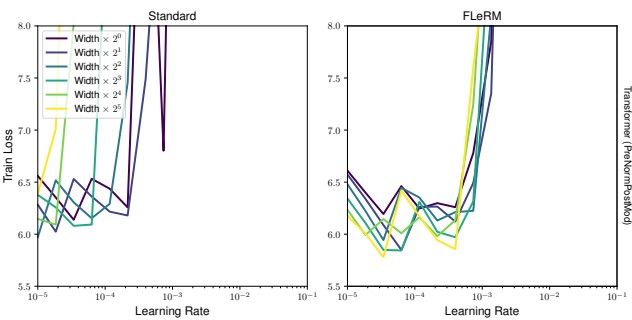

*Figure 20.* Width transfer plot for Transformer (PreNormPostMod) using **SGD instead of Adam**. Momentum 0.9.

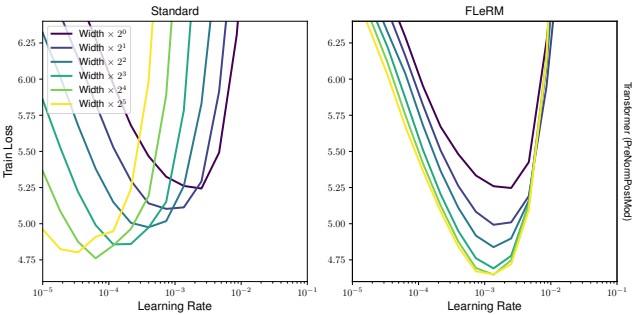

*Figure 21.* Width transfer plot for Transformer (PreNormPostMod) using **SignSGD / Signum instead of Adam**. Momentum 0.9.

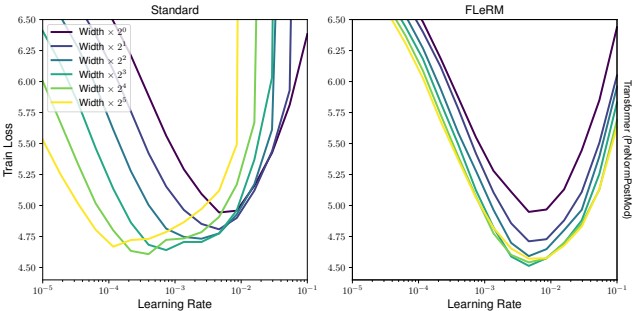

*Figure 22.* Width transfer plot for Transformer (PreNormPostMod) using **AdamW instead of Adam**. Weight decay 0.1.

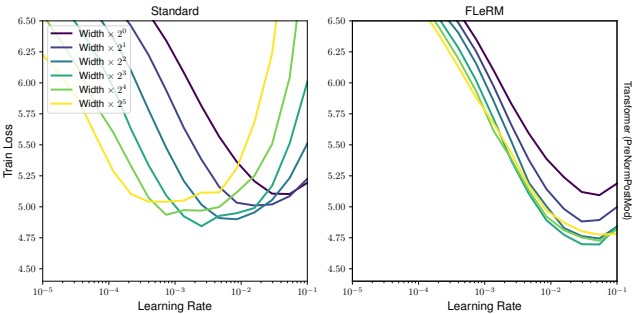

*Figure 23.* Width transfer plot for Transformer (PreNormPostMod) using **Adamax instead of Adam**.

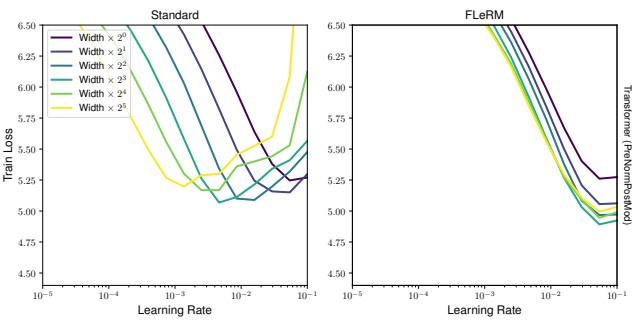

*Figure 24.* Width transfer plot for Transformer (PreNormPostMod) using **Adagrad instead of Adam**.

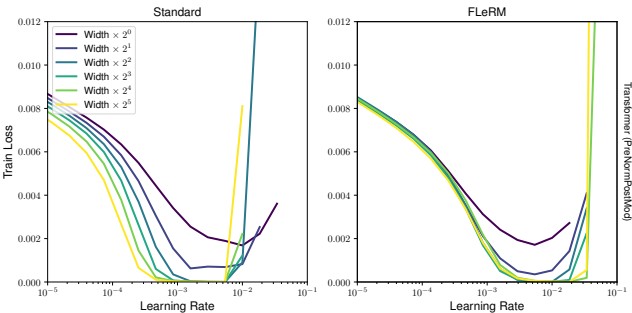

*Figure 25.* Width transfer plot for ResMLP using **SGD instead of Adam**. Momentum 0.9.

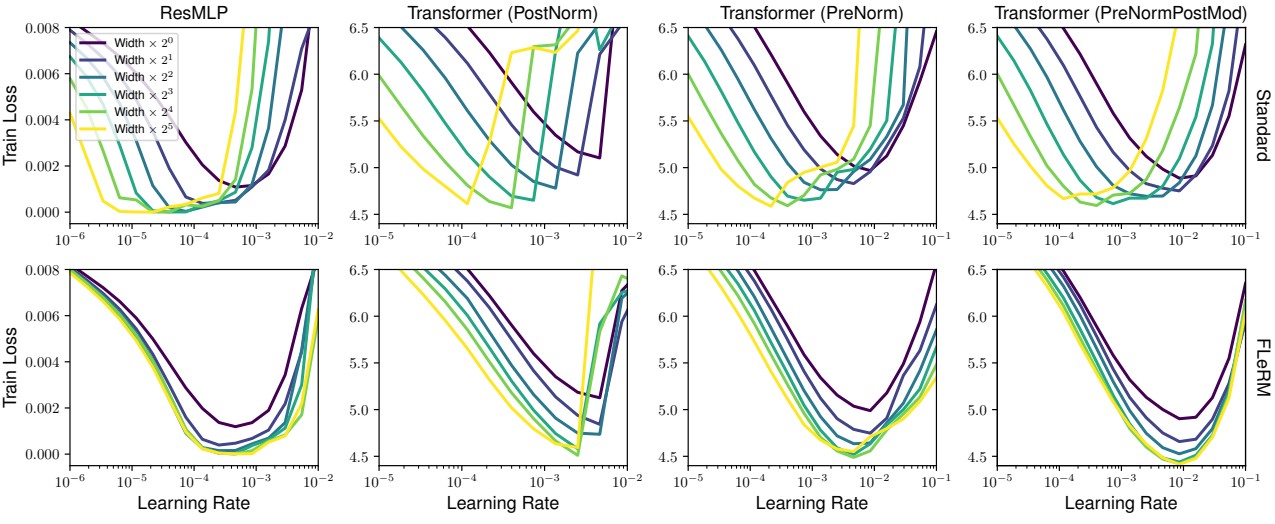

*Figure 26.* Width transfer experiments as in main paper, but using output independence tricks given in Section D

