# OpenReview forum: "Function-Space Learning Rates"
_ICML.cc/2025/Conference — ICML 2025 poster_

### Official Review · Reviewer_ALy5 · 2025-03-10

**Overall Recommendation:** 3

**Summary:**

This paper introduces FLeRM, an optimization algorithm that trains a smaller model, records function-space learning rates, and uses those recorded learning rate in the training process of a larger model.

The authors perform experiments on various datasets such as CIFAR-10 and Wikitext-103. They use residual MLP and Transformer architectures for their experiments. They show that using the proposed method reduces the effect of different learning rates on the training loss. They also, explore the effect of increasing the depth and width of neural networks and the learning rate adjustments necessary when such changes are made.

## Update after rebuttal

I wish to thank the authors for the rebuttal and the reviewers for their thoughtful comments.

The issue of dataset size and pertinence to modern deep learning landscape is an important one. I understand that during the rebuttal timeframe it is not feasible to run additional experiments. Regardless, it is an issue that needs to be addressed.

I have decided to keep my original rating.

**Claims And Evidence:**

There is a serious issue with the proposed algorithm that poses a major limitation on the usability of the algorithm in real-life modern optimization scenarios. The main claim of this paper is the use of a smaller network to optimizer a larger one. This would reduce the amount of computation required for the larger network. However, according to Algorithm 1, the proposed algorithm needs to record and store the model parameters, $W^l$, at every iteration $t$.
This would require double the amount of memory to maintain this buffer. This becomes problematic when training large models, such as LLMs that already require multiple GPUs for training.

**Essential References Not Discussed:**

NA

**Experimental Designs Or Analyses:**

1. Throughout the paper, the authors present and compare training loss for their method and the vanilla approach. While training loss can be an indicator of learning, validation/test loss is required to see if their method improves generalizability (All figures, including the appendix, focus on the training loss). Generalizability is the main focus of optimization. It is possible that lower training loss leads to higher validation loss.

2. In Figure 3, the authors show the values of the training loss for vanilla and the proposed optimization scheme. In various cases, the best training loss achieved by vanilla methods is better (e.g., ResMLP). There is no advantage of less variance to learning rates when the final training loss is worse using the proposed method.

**Methods And Evaluation Criteria:**

The datasets used (e.g., CIFAR-10 and Wikitext-103) are good starting points for the paper. However, in the modern deep learning landscape, these smaller datasets are less relevant. Datasets such as ImageNet have become the bare minimum to effectively examine the efficacy of new optimization approaches.

**Other Comments Or Suggestions:**

NA

**Other Strengths And Weaknesses:**

- Without results on validation/test set data, the findings of the paper have limited significance for the academic and practical use cases. Please refer to the comments in **Experimental Designs Or Analyses** for more details.

**Questions For Authors:**

1. Do you have comparative results on the memory and computation requirements of your method versus vanilla approaches that would put into perspective the extent of the limitations of your approach?

2. Why did you decide not to focus on validation/test set loss, instead focus on the training loss?

3. If your method is applied to a larger, noisier dataset, such as ImageNet, what potential issues do you expect to see with your approach?

**Relation To Broader Scientific Literature:**

Hyperparameter transfer has been focus of research for a long time. This paper proposes using learning rates found in a smaller network and apply them to a larger network to improve invariance to hyperparameters.

**Theoretical Claims:**

Yes, all of them.

---

> ### Author Rebuttal · Authors · 2025-04-01
>
> Thank you for your positive and thoughtful review!
>
> > ...the proposed algorithm needs to record and store the model parameters at every iteration. This would require double the amount of memory to maintain this buffer...
>
> We agree that if we needed to store an extra copy of the weights at every step, that would be bad.  We definitely don't do that! Specifically, the extra copy of the weights is only necessary at at the time steps where you use FLeRM to compute the layerwise learning rates.  And we do this very infrequently.  For example, in the FLeRM experiments in the main paper, we apply FLeRM at the start and then fix those layerwise learning rates for the rest of training.  After that, training requires only the same time and space as standard training: the only difference is the layerwise learning rates.
>
> Moreover, note that an extra copy of the weights wouldn't increase peak memory usage by nearly as much as a factor of 2.
> That's because Adam already stores not only the weights themselves, but also the average gradients and squared gradients, and because backprop requires storing a large number of intermediate activations.
>
> We agree that the peak memory usage during a FLeRM step is higher due to this extra copy of the weights, but as it happens so infrequently, there are a number of strategies to mitigate the issue, including using smaller batch sizes for a FLeRM step, or shifting some quantities to CPU memory.
>
> > ...computation requirements of your method versus vanilla approaches...
>
> For the width experiments in the main text we found that FLeRM increased the runtime by around 1.7\%, which is negligible relative to the benefits achievable by accurate hyperparameter transfer.
>
> > these smaller datasets are less relevant...
>
> We agree that it would be ideal to be able to use larger datasets.  However, bear in mind that our datasets are common across previous hyperparameter transfer works, e.g. MuP [1], and that for each panel in the hyperparameter plots, we must train the network dozens of times (for every learning rate, for every width / depth etc.) and so very large scale pretraining tasks are not practical, even given access to a reasonable amount of compute.
>
> > ...validation/test loss is required to see if their method improves generalizability (All figures, including the appendix, focus on the training loss)... Why did you decide not to focus on validation/test set loss, instead focus on the training loss?
>
> We plotted the test loss for the Transformer (PreNormPostMod) width transfer experiment in [Rebuttal Figure 2](https://fslricml2025rebuttalsfigures.tiiny.site) (hyperlink to anonymous url), which showed exactly the same patterns as the train loss. (Using the train-loss is common in the hyperparameter transfer literature [1,2,3] because in the LLM-pretraining setting where you usually train on each datapoint only once, the expected train and validation loss turn out to be equivalent [4]).
>
> > In various cases, the best training loss achieved by vanilla methods is better (e.g., ResMLP).
>
> Sometimes the loss is better with FLeRM.  This is most clear in:
> * Figure 2 (main paper): PreNorm, PreNormPostMod
> * Figure 3 (main paper): PostNorm, PreNorm
> * most of the [new figures we have done for the Rebuttal](https://fslricml2025rebuttalsfigures.tiiny.site).
>
> We agree though that sometimes the loss is better without FLeRM.  This is most clear in:
> Figure 3 (main paper): ResMLP, PreNormPostMod
>
> If anything, we would argue that when FLeRM seems to better in more relevant settings (especially PreNorm Transformers). Importantly though, we did not motivate our work as improving performance so we did not perform the large-scale experiments necessary to definitively establish performance improvements.
> Nonetheless, this is definitely a super-exciting direction for future work.
>
> > If your method is applied to a larger, noisier dataset, such as ImageNet, what potential issues do you expect to see with your approach?
>
> Hyperparameter scaling work e.g. [1,2,3] has not, to our knowledge, found qualitative differences in scaling across different datasets, so we believe we are unlikely to see any such differences here either.
>
> [1] Yang, G., Hu, E. J., Babuschkin, I., Sidor, S., Liu, X., Farhi, D., Ryder, N., Pachocki, J., Chen, W., and Gao, J. Tensor programs v: Tuning large neural networks via zero-shot hyperparameter transfer, 2022
>
> [2] Yang, G., Yu, D., Zhu, C., and Hayou, S. Tensor pro- grams vi: Feature learning in infinite-depth neural net- works, 2023.
>
> [3] Bordelon, B., Noci, L., Li, M. B., Hanin, B., and Pehle- van, C. Depthwise hyperparameter transfer in resid- ual networks: Dynamics and scaling limit, 2023
>
> [4] Aitchison L. Why you don't overfit, and don't need Bayes if you only train for one epoch. arXiv preprint arXiv:2411.14478. 2024 Nov 19.

---

### Official Review · Reviewer_JVrs · 2025-03-12

**Overall Recommendation:** 4

**Summary:**

This paper provides a novel method to transfer learning rates across model sizes. The approach is very flexible as it leverages monte-carlo estimation of the changes in model outputs under a proposed change in one of the weight matrices. The authors show that their approach can enable consistent optimal learning rates while training models of different widths and depths as well as LORA rank during fine tuning. They also show the ability of their algorithm to transfer across initialization scale. The flexibility of the approach makes it especially attractive.

**Claims And Evidence:**

The theoretical results are supported by proofs and derivations and the entire approach is supported by a large array of experiments.

**Essential References Not Discussed:**

N/A

**Experimental Designs Or Analyses:**

Yes, the experimental design is reasonable in my opinion.

**Methods And Evaluation Criteria:**

Yes, the experiments seem reasonable. They looked at ResNets on CIFAR, transformer models on Wikitext 103, and finetuning language models on Cold French Law and Mathpile.

**Other Comments Or Suggestions:**

1. In equation 16 it should be $i, i'$ in the sum not $i i$. There are also two equal signs in 16.
2. Blue line in algorithm should be flipped. I think it should be $|\Delta f_{base}| / |\Delta f|$ instead of $|\Delta f | / |\Delta f_{base}|$5.
3. The authors mention the limitations of scaling theory by citing Everett et al and the fact that weight alignment to input features in hidden layers are dynamic over time. I would like to point out that **even in the infinite limit** we would not expect perfect alignment $A=1$ which would require the feature vectors to become singular vectors of the weight matrix and there would be complex dynamics for $A(t)$. Concretely, with SGD a weight matrix would have the form $W(t) = W(0) + \frac{\eta}{N} \sum_{t} g_t \phi(h_t)^\top$ where $g_i$ and $\phi_j$ are vectors with $O(1)$ entries. If I pass the last vector $\phi(h_T)$ through this matrix I get $W(0) \phi(h_T) + \eta \sum_{t} g_t  \left( \frac{1}{N} \phi(h_t) \cdot \phi(h_T) \right)$. In general, $\phi_T$ is not a singular vector of $W(t)$ so the alignment will be lower than one and also changing over time in a way that depends on the correlation structure of the $\phi$'s. Thus in my understanding their experiments do not really invalidate the scaling theory to the extent they claim.

**Other Strengths And Weaknesses:**

The paper provides an interpretable algorithm and several useful experiments. One potential drawback is the need to train a side by side base model to keep track of the rescaled learning rates for each layer. In addition, if the base model has many layers there are potentially many scalars to track, but this is still pretty cheap.

**Questions For Authors:**

1. Do the authors have a sense of how the quality of their estimate degrades with either (A) the length of steps / training time between learning rate adjustments (the value set to 100 in pseudocode) or (B) the number of monte carlo samples? Since only a small number of scalars need to be estimated to control the norm, it seems likely that very few samples are needed. However, I wonder if longer delays between updating  could really impact performances of models which have been scaled up significantly compared to the base model. Are there any experiments with this?
2. On depth scaling to a model with $L$ blocks, the authors rescale the base function LR $|\Delta f_{base}| \to \frac{1}{L}$ and introduce branch scale factors $1/\sqrt{L}$. I believe that this would lead to $\eta_L = \eta_0$ for SGD and $\eta_L = \eta_0 / \sqrt{L}$ for Adam. Both of these match the scaling theory for $1/\sqrt{L}$ branch scaling. However, there are other possible depth scalings. For example, if one adopts a $1/L$ branch scale, the residual blocks do not linearize in the limit and there is within-block feature learning as $L \to \infty$ (see section 3.4 here https://arxiv.org/abs/2405.15712).
3. Do the authors have a sense of why their experiments do not always show improved performance with respect to model size? Deeper networks seem to be worse at their optimal learning rates in some settings.
4. Do the authors think their approach is mathematically similar to controlling the scale of the instantaneous NTK?
$$df \sim \sum_{\ell} \eta_\ell \frac{\partial f}{\partial W_{\ell} } \cdot \frac{\partial f}{\partial W_{\ell}}$$

**Relation To Broader Scientific Literature:**

This paper is studying an important problem in the science of scaling neural networks (how to make their optimization profiles consistent across model sizes). It introduces a novel and flexible approach that would be easy and cheap to implement for the practitioner.

**Theoretical Claims:**

The proofs and derivations are correct and straightforward from my reading.

---

> ### Author Rebuttal · Authors · 2025-04-01
>
> Thank you for your positive review noting that we introduce "a novel and flexible approach that would be easy and cheap to implement for the practitioner."
>
> > One potential drawback is the need to train a side by side base model...
>
> This is unavoidable whenever doing any form of hyperparameter transfer, as hyperparameter transfer by definition involves tuning the hyperparameters on a small base model, then using some strategy (e.g. muP) to transfer those hyperparameters to the scaled model.
>
> > potentially many scalars to track, but this is still pretty cheap.
>
> Yes, our Kronecker scheme only needs $LD$ scalars if we have $L$ tensors with $D$ dimensions each, which is insignificant compared to the costs of training the network itself.
>
> > In equation 16 it should be $i,i'$ in the sum not $ii$. There are also two equal signs in 16.
>
> > Blue line in algorithm should be flipped...
>
> Thanks! Fixed!
>
> > The authors mention the limitations of scaling theory by citing Everett et al...
>
> Thanks, this is very interesting! We have updated this section to emphasize that even in theory in the infinite-width limit, alignment is potentially very complex, with values lower than $1$ that can change over time. Do let us know if there are any references we should cite on this point. In any case, our point here is merely that deriving the "correct" alignment is very complicated, which suggests that our more "empirically led" approach is a useful alternative.  For the purposes of that point, it doesn't really matter whether the complexity can in-principle be captured in the infinite-width limit, or only emerges empirically.
>
> > Do the authors have a sense of how the quality of their estimate degrades with either (A) the length of steps / training time between learning rate adjustments (the value set to 100 in pseudocode) or (B) the number of monte carlo samples?
>
> In the main paper's hyperparameter transfer experiments, we only adjust the learning rate at initialisation, and keep it constant for the rest of training. In Appendix C.2, we did it every 100 steps (as in the algorithm). We found very little difference between these two settings, suggesting that hyperparameter transfer mainly depends on correcting "constant" differences in function-space learning rates, rather than varying wildly throughout training. We did compare the bias and variance of the estimator using different covariance assumptions: See [Rebuttal Figure 1](https://fslricml2025rebuttalsfigures.tiiny.site) (hyperlink to anonymous url).
>
> > Depth scaling
>
> This is super interesting.
> Certainly, we agree with [1] that you want changes in the attention patterns to be constant as you scale depth, and that isn't at all trivial to achieve (as the function-space learning rate for $W_K$ and $W_Q$ is basically the change in the output of attention, multiplied by the init for $W_V$ and $W_O$).
> As such, we're pretty sure we agree that with FLeRM, changes in the attention would vanish in the infinite depth limit using the $1/sqrt(L)$ init, but not using the $1/L$ init.
> We'll have to think about the right way to handle this in the FLeRM setting, to be robust to e.g.\ different choices of normalization and changes over time during training, but we're confident that FLeRM has enough flexibility to handle it correctly.
> One interesting approach would be to apply FLeRM to e.g. the self-attention layer with randomized $W_V$, rather than just to the overall network output.
> That would allow you to isolate the change in the attention patterns, and ensure they remained constant as you scale depth.
> But we will definitely have to think harder about it!
>
> [1] https://arxiv.org/abs/2405.15712
>
> > Do the authors have a sense of why their experiments do not always show improved performance with respect to model size?
>
> This is a very interesting phenomenon. We speculate that as models get larger in the "standard" setting, there are shifts in the relative size of the function space learning rates for different parameters, and that sometimes these changes are actually beneficial to performance.
> FLeRM, by fixing the function-space learning rates to those in the base model, might eliminate some of these beneficial changes.
> We're super-excited to pursue follow-up work which uses function-space learning rates to investigate some of these phenomena in-depth.
>
> > Do the authors think their approach is mathematically similar to controlling the scale of the instantaneous NTK?
>
> There likely is a connection, and this would be an interesting direction to explore in the future.

---

> > ### Comment · Reviewer_JVrs · 2025-04-03
> >
> > I thank the authors for their detailed responses. I will maintain my score.

---

### Official Review · Reviewer_fHgH · 2025-03-13

**Overall Recommendation:** 4

**Summary:**

The paper defines function space learning rate as the rate of change of a neural network's outputs per training iteration. Then, the method FLeRM is introduced for either estimating the per-layer function space learning rate of a model over training, or setting these learning rates (LRs) to fit an arbitrary schedule. Finally, experiments determine if the method can enable LR transfer between smaller width and/or depth networks and larger networks, as well as LoRA adapter LRs, by first recording and then setting per-layer function space learning rates.

**Claims And Evidence:**

Claim: FLeRM efficiently estimates function-space LRs. True - although there are some caveats (see Methods below).

Claim: FleRM enables LR transfer between networks of different width, depth and LoRA adapters of different rank. True - although evidence could be strengthened (see Experimental Design and Questions below).

**Essential References Not Discussed:**

I am not aware of any missing references.

**Experimental Designs Or Analyses:**

The experiments and analyses are sound, albeit not sufficiently large-scale to demonstrate transfer in cutting-edge models (although it is fair to say this would be out of scope). Some of the model architecture choices are a bit strange - see Questions below.

**Methods And Evaluation Criteria:**

The method is elegant but could also be overcomplicating things. The authors say equation (1) is intractable, which is reasonable for large models/datasets. However, one can consider a simpler Monte Carlo estimator of (1) that is just the change in output between two iterations, computed over some minibatch. One could even schedule the same minibatch close together in training so as to get this information "for free" during training. Another (speculative) possibility is to take the change in loss between successive iterations, and estimate change in output from the derivative of loss with respect to output. In any case, the authors should justify why their method is preferred over some other estimator (via variance of the estimator, time complexity, etc.) - see Questions (below) for more on this.

Some other assumptions which are reasonable, but could use some empirical support are:
1. What is the empirical performance of the method if one assumes $Cov[Z_{ij}, Z_{i'j'}]$ is diagonal or full rank? While the latter is probably impractical, the former is an even simpler assumption than the covariance being factorizable, and is also more closely related to existing work (e.g. Adam) - thus it would be helpful to have a comparison of the proposed method versus this simpler approach.
2. Learning rate sweeps are for global learning rates, but this method already sets learning rates for individual layers. Thus, although costly, it would be informative to do per-layer learning rate sweeps (perhaps via random or grid search) for at least some small settings, as well as per-layer learning rate transfers. Transferring LRs found via per-layer sweeps on small models might even be a way to cheaply improve performance on deeper models.
3. Related to point 2, another assumption is to share the learning rate over multiple layers when transferring to a deeper model. Although appendix C.4 conducts the ablation of dividing learning rates equally across layers, if point 2 is addressed by finding optimal per-layer learning rates, seeing if those rates are uniform over blocks would strengthen this assumption.

**Other Comments Or Suggestions:**

Some comments on notation in section 3:
- $d$ should be $\partial$ in equations 1, 4, 5, 8.
- it would be helpful to give the dimensions of the matrices $Z_{ij}$, $U$, and $V$.
- $||\Delta_l f||^{base}_{RMS}||$ should be defined

Other presentation issues:
- line 301: "as suggested *in* Section 3.2"
- Figure 1 and 2 are too far removed from the discussions. Also, the grid lines could be stronger and the plotted lines thinner (it is hard to tell which direction the trends are in due to the large number of layers being plotted).
- Algorithm 1 is also separated by several pages from its discussion
- Some claims are made about minor improvements in performance - adding a table of test loss would make this easier to evaluate than comparing between figures.

**Other Strengths And Weaknesses:**

Strengths: as discussed above, the method is complementary to existing theoretically oriented work around learning rates and parameter scales. It is also useful both as a tool for LR selection, and as a tool for analyzing training dynamics. The paper is really well presented and the experiments are thorough. I am particularly looking forward to the possibility of empirically measuring output dynamics and relating them to various theoretical predictions.

Weaknesses: as discussed above, the method might be needlessly complicated. The experiments also do not take full advantage of the method's potential and so I am unsure how much significance the results have. If the results cannot extend beyond the slightly non-standard settings explored by the paper, and beyond rescaling per-layer learning rates by constants, then the impact is somewhat limited. If the results do generalize to more settings and more complex learning rate schedules, then the work is very significant.

## Update after rebuttal

The authors have answered all of my questions, and I stand by my review that this paper has significant contributions and should be accepted.

**Questions For Authors:**

What is the variance of the proposed estimator? What is the magnitude of the bias introduced by assumptions on the covariance matrix? What is the time complexity of the proposed estimator? Either derivations or empirical evaluations (e.g. comparing the method to the naive approach of estimating equation 1) are welcome.

Could the authors discuss the advantages/disadvantages of considering changes in the function output as opposed to loss? I can see some advantages (e.g. function output is not sensitive to choice of loss), but for the sake of argument, why not look at $|| \Delta_l L ||_2^2$ instead, where $L$ is the loss function?

Why ResMLP instead of a convolutional ResNet?, Also, why is Layernorm/Batchnorm not used in the ResMLPs? Similarly, why disable affine transformations in the transformer Layernorms?

**Relation To Broader Scientific Literature:**

The empirical-first approach to measure the relationship between weight changes and function space changes is complementary to existing theoretical-first approaches. Not only is this method useful for its stated purpose (hyperparameter selection and transfer), but as demonstrated in figure 1, it could also be a way to generate empirical evidence on training dynamics. This would be a useful tool in literature that looks at how outputs evolve due to changes in weight space (e.g. neural tangent kernel literature, Lipschitz-based complexity bounds).

**Theoretical Claims:**

The derivation of the estimator in section 3.1-3.2 appears sound. Some properties of the estimator are not addressed (see Questions below).

---

> ### Author Rebuttal · Authors · 2025-04-01
>
> Thank you for your review, stating that "The paper is really well presented and the experiments are thorough. I am particularly looking forward to the possibility of empirically measuring output dynamics and relating them to various theoretical predictions." (We're looking forward to that too!)
>
>
> > one can consider a simpler Monte Carlo estimator... Another possibility... estimate change in output from the derivative of loss with respect to output.
>
> Our method is unique in that, from a single forward+backward pass, it returns $L$ quantities where $L$ is the number of parameter tensors.
> These $L$ quantities are the change in the function induced by the change in only one specific parameter. In contrast, finite differences based approaches like the one you suggest will only tell us how rapidly the outputs are changing overall, not how much individual layers contribute to this change.
> You could of course compute how individual layers contribute using finite differences, but that would require $L$ forward passes, where, in each forward pass you perturb only one of the $L$ parameter tensors.
>
> > Could the authors discuss the advantages/disadvantages of considering changes in the function output as opposed to loss?
>
> Considering the change in output, rather than the loss is the usual approach taken e.g. in muP [2] and Modula [3].  In our preliminary experiments, we did try forcing the loss to change by a specific amount, but we found training to be unstable as the loss approached the minimum. This is perhaps expected: once the loss is at the minimum, it can't go down any further, so trying to force the loss to go down a specific amount further is not sensible, and you might expect it to lead to instability.
>
> > What is the empirical performance of the method if one assumes is diagonal or full rank? ... what is the variance of the proposed estimator? What is the magnitude of the bias introduced by assumptions on the covariance matrix?
>
> We have included a plot comparing the bias and variance of different covariance assumptions in the appendix. See [Rebuttal Figure 1](https://fslricml2025rebuttalsfigures.tiiny.site) (hyperlink to anonymous url).
>
> > it would be informative to do per-layer learning rate sweeps
>
> > another assumption is to share the learning rate over multiple layers when transferring to a deeper model.
>
> This would be a super-interesting question for follow-up work!
> Of course, it is not in scope for the present paper due to the very extensive new experiments required, along with the careful thought required to draw conclusions from those experiments.
> That said, we do hope that our notion of function space learning rates would help design these sweeps efficiently.
>
> > I reviewed appendix A. Appendix A should include the details of the training task for ResMLP.
>
> Fixed.
>
>
> > Regarding appendix C.2 - when updating the LR at time t, is it made to match the base model's LR at time 0, or time t?
>
> At time t! Appendix C.2 is "change the per-layer learning rates every 100 iterations to make the function-space learning rates match the base model at that time point", whilst the main paper experiments are "change the per-layer learning rates at the beginning of training to make the function-space learning rates match the base model at initialisation, then use those learning rates for the rest of training". Our experiments seem to suggest that, for the purposes for hyperparameter transfer, it is sufficient to "correct" the learning rate at initialisation.
>
> Other comments and suggestions: All fixed.  Thanks!
>
> > What is the time complexity of the proposed estimator?
>
> For the width experiments in the main text we found that FLeRM increased the runtime by around 1.7\%, which is negligible relative to the benefits achievable by accurate hyperparameter transfer.
>
> > Why ResMLP instead of a convolutional ResNet?, Also, why is Layernorm/Batchnorm not used in the ResMLPs?
>
> The ResMLP serves as an extremely simple setting, and a similar architecture is used in previous work of depthwise hyperparameter transfer [1].
>
> > why disable affine transformations in the transformer Layernorms?
>
> We disabled affine transformations early on in the project to reduce complexity in prototyping. We have rerun the Transformer (PreResPostMod) width transfer experiment with affine transformations enabled and found no problems. See [Rebuttal Figure 4](https://fslricml2025rebuttalsfigures.tiiny.site).
>
> [1] Yang, G., Yu, D., Zhu, C., and Hayou, S. Tensor programs vi: Feature learning in infinite-depth neural networks, 2023.
>
> [2] Yang, G., Hu, E. J., Babuschkin, I., Sidor, S., Liu, X., Farhi,
> D., Ryder, N., Pachocki, J., Chen, W., and Gao, J. Tensor
> programs v: Tuning large neural networks via zero-shot
> hyperparameter transfer, 2022
>
> [3] Bernstein, J. and Newhouse, L. Modular duality in deep
> learning, 2024a

---

### Official Review · Reviewer_pf7i · 2025-03-14

**Overall Recommendation:** 3

**Summary:**

This paper introduces the concept of function-space learning rates, which measure the magnitude of changes in a neural network's output function in response to updates in parameter space. The authors propose an efficient Monte-Carlo-based method to estimate these function-space learning rates and introduce FLeRM, a technique designed to transfer hyperparameters from smaller "base" models to larger ones as updates in function space are scale invariance. The effectiveness of FLeRM is demonstrated empirically through multilayer perceptrons (MLPs) and transformer architectures, enabling the transfer of the optimal learning rate across width, depth, initialisation scale, and LoRA rank.

**Claims And Evidence:**

The claims are generally supported with convincing evidence. The claim that FLeRM can be used for hyperparameter transfer in large-scale LLM training is not convincingly supported, as the tested models are significantly smaller (millions of parameters) than foundational LLMs (hundreds of billions to trillions of parameters).

**Essential References Not Discussed:**

All essential references to my knowledge are discussed.

**Experimental Designs Or Analyses:**

The experimental design appears valid.

**Methods And Evaluation Criteria:**

The methods are extensively tested on a number of different experimental setups. See the weakness section for additional experiments to support your analysis.

**Other Comments Or Suggestions:**

1.	The learning rate update in Algorithm 1 (line 138) seems to contradict the theoretical analysis. I believe the numerator and denominator should be in reverse order.
2.	Minor typo in line 261 – missing a closing bracket in x+f(Norm(x)

**Other Strengths And Weaknesses:**

Strengths
1.	The manuscript presents an innovative way to understand neural network training dynamics by shifting the focus from parameter-space learning rates to function-space learning rates.
2.	The proposed Monte Carlo-based method, combined with Kronecker factorization, enables the estimation of function-space learning rates with minimal computational overhead, by requiring only a single additional forward and backward pass.
3.	The paper introduces FLeRM, a robust solution for hyperparameter transfer that can be used with any network architecture and at any point during training.
4.	The paper extensively evaluates FLeRM in multiple scenarios, including scaling width and depth in MLPs and transformers, varying initialization scales, and adapting LoRA rank, demonstrating its versatility.

Weaknesses
1.	The paper evaluates FLeRM primarily with the Adam optimizer. It would be beneficial to compare it with other optimizers such as SGD and AdamW to establish its robustness across different optimization paradigms.
2.	While FLeRM is shown to be effective for width and depth scaling separately, real-world scaling typically involves increasing both simultaneously. Showing results for this experimental setting will be helpful.
3.	The experiments seem to be limited to constant learning rate schedules. Does FLeRM also work with dynamic LR schedules?
4.	While Appendix A provides model details, explicitly stating the exact parameter counts and layer configurations across different width and depth settings in the main text would help clarify the extent of scaling in the experiments.
5.	Despite strong empirical results, the paper lacks rigorous theoretical analysis or formal guarantees regarding the optimality and convergence properties of function-space learning rates. As such, the authors claim that FLeRM could facilitate hyperparameter transfer in large-scale LLM training is not convincingly supported, as the tested models are significantly smaller (millions of parameters) than foundational LLMs (hundreds of billions to trillions of parameters).

**Questions For Authors:**

See the weakness section

**Relation To Broader Scientific Literature:**

The paper’s approach in measuring and setting function space learning appears to be unique. I am not aware of any other papers in this area. In terms of hyperparameter transfer, the paper extends the current literature by removing some of the rigid assumptions about initialisation and the need to set a number of hyperparameters in existing methods.

**Theoretical Claims:**

The theoretical claims of the paper appear correct.

---

> ### Author Rebuttal · Authors · 2025-04-01
>
> Thank you for your thoughtful review with many excellent suggestions, acknowledging that "3. The paper introduces FLeRM, a robust solution for hyperparameter transfer that can be used with any network architecture and at any point during training. 4. The paper extensively evaluates FLeRM in multiple scenarios, including scaling width and depth in MLPs and transformers, varying initialization scales, and adapting LoRA rank, demonstrating its versatility."
>
> We have added the additional experiments your requested (see below).
>
> ### 1) Other optimisers
>
> We have run the Transformer (PreNormPostMod) width transfer experiment using SGD, signSGD, AdamW, AdamMax, Adagrad instead of Adam. Please see [Rebuttal Figures 6-10](https://fslricml2025rebuttalsfigures.tiiny.site) (hyperlink to anonymous url). As before, FLeRM aligns the optimal learning rates, and in this case even improves the best loss achieved. There is instability when training the transformer with SGD (SGD is not often used for transformers for this reason). We also tried SGD on ResMLP in [Rebuttal Figure 11](https://fslricml2025rebuttalsfigures.tiiny.site), which worked fine.
>
> ### 2) Scaling width and depth simultaneously
>
> We have run the Transformer (PreNormPostMod) experiment scaling both width and depth simultaneously (up to 8x because of computational constraints) and observed hyperparameter transfer. Please see the [Rebuttal Figure 5](https://fslricml2025rebuttalsfigures.tiiny.site).
>
> ### 3) LR scheduling
>
> We have run the Transformer (PreNormPostMod) width transfer experiment with the CosineAnnealingLR scheduler and observed hyperparameter transfer. Please see the [Rebuttal Figure 3](https://fslricml2025rebuttalsfigures.tiiny.site).
>
> Note that the limited time available means we haven't been able to run the full range of models from the original manuscript for all of these new settings. We will run a systematic sweep for the camera ready. We hope you agree that the results we do have indicate it is unlikely that there will be any surprises in the final figures.
>
>
> ### 4/5 Model size
>
> We have added details on model sizes to the manuscript.  Specifically, the widest model (in the width scaling plot in Figure 2) we considered contains 814M parameters. We agree that this is of course far smaller than the very largest modern LLMs.  But at the same time, it isn't so small (e.g. there is alot of interest at the moment in training ~1B parameter reasoning models). Please also remember that FLeRM forms only one part of our contribution, with our main contribution being the efficient estimate of layerwise function-space learning rates, which has many possible uses, including analysing training dynamics (as shown in Section 4.1) and hyperparameter transfer (FLeRM, section 4.2).
>
>
> > The learning rate update in Algorithm 1 (line 138) seems to contradict the theoretical analysis. I believe the numerator and denominator should be in reverse order.
>
> Thanks! Fixed!
>
> > Minor typo in line 261 – missing a closing bracket in x+f(Norm(x)
>
> Thanks! Fixed!
>
> ### Conclusions
>
> Thank you for generously outlining the paper's strengths in your original review. We hope this response (and especially the new experimental results) have addressed your key concerns. If so, we would greatly appreciate it if you would reconsider your score.

---

> > ### Comment · Reviewer_pf7i · 2025-04-04
> >
> > The authors have justified their approach and I am happy to upgrade my evaluation to Weak Accept.

---

### Decision · Program_Chairs · 2025-05-01

**Decision:**

Accept (poster)

**Comment:**

The paper introduces a method for estimating the sensitivity of the output of a neural network with respect to the parameters. The method estimates this sensitivity using a monte carlo estimate, together with an assumed kronecker structure of the covariance over elements of the layer. The resulting sensitivity now depends on three scalars per layer (or group of layers), which are estimated in an online fashion using an exponential moving average. The overall idea is new, interesting, and the experiments also validate the approach. Since the reviewers were also positive, I recommend the paper be accepted. But I do see one thing I would think would improve the paper, and that is
- [Baseline comparison] The main application for your technique is hyperparameter transfer across model scales. Yet you do not compare to any baseline for this task, such as the muP or modula. I understand that comparing to modula can be difficult, since this would require re-writing your models in modula. But could you not at least show how your method compares to muP in practice? This would really help to get the attention of practitioners.

I have also some smaller recommendations:

- [Misleading title for theoreticians] For theoretical subcomunities of ICML, the title could be misleading. Function space learning is sometimes used to refer to learning the model over the space of functions, perhaps by lifting to the space of measures. It would typically make theoretical readers think that you are considering optimization analysis in infinite-dimensional function spaces, such as Functional gradient descent (like boosting or kernel methods). I would recommend renaming the paper to more precisely target your audience, for instance
"Learning the sensitivity of a function online, with an application to hyperparameter transfer"
Or something that conveys the same meaning. This is just a recommendation.

- [Relationship between sensitivity and learning rates] I would recommend expanding on why you use the relative change in the sensitivities of the big model to the base models to set the learning rate in (19). This is somewhat intuitive: more sensitive models should have smaller learning rates. But since this is key to how you apply your technique, it would be best to clarify further.

- Because you initialize your momentum buffers at zero EMA Z2[ℓ], EMA ZZT[ℓ], EMA ZTZ[ℓ] = 0, the resulting EMA estimates are not discrete expectations over the past samples. You might glean some minor improvements by correcting for this bias. For this you would need to either use the "bias correction" trick of Adam [Adam], or initialize these buffers at the first sample, instead of zero [Section 2.3, Schaipp2024]. Either one of these approaches will avoid the issue of your momentum buffer starting too small.



[Adam] Kingma, D. P., & Ba, J. (2015). Adam: A Method for Stochastic Optimization. In Proceedings of the 3rd International Conference on Learning Representations (ICLR).

[Schaipp2024]  Fabian Schaipp, Ruben Ohana, Michael Eickenberg, Aaron Defazio, Robert M. Gower, MoMo: Momentum Models for Adaptive Learning Rates, ICLR 2024.